# The syntax of wh-questions in unaccusative and (Un)ergative structures in Mehri language: A Phase-based approach

**Abdul-Hafeed Fakih**[1], **Saeed Saad Al-Qumairi**[2], **Ali Abbas Falah Alzubi**[1]*

**1** Department of English, College of Languages and Translation, Najran University, Najran, Saudi Arabia,
**2** Faculty of Education, Mahrah University, Mahrah, Yemen

* Aliyarmouk2004@gmail.com

**Data Availability Statement:** All data is available within the paper.

**Funding:** Yes, this work was funded by the Deanship of Scientific Research at Najran

## Abstract

The Mehri Language is an endangered language spoken in eastern Yemen, a sub-group of the Semitic language family, and a Southern Arabic language. The syntax of Mehri wh-questions has not been explored within minimalism; hence, there is a morpho-syntactic need to provide a modern analysis of wh-questions in order to show how the interrogative structures can be derived. This study aims to examine the syntax of the wh-question movement in Mehri's unaccusative/ergative and unergative structures and answer the following questions within Chomsky's (2000 and 2008) Phase-based Theory: (i) Does the Mehri language allow fronting of wh-phrases to [Spec-CP]? And (ii) how can wh-movement in Mehri unaccusative and (un)ergative structures be accounted for? This work presents a novel analysis of wh-question movement in unaccusative/ergative and unergative structures in Mehri; it demonstrates that the source head C triggers the movement of wh-adjunct and wh-subject phrases. In wh-adjunct extraction, two strategies are employed: overt wh-fronting and wh-in-situ; when the head Foc inherits an edge feature from C, wh-adjunct overtly undergoes movement from its original position within v*P to the left peripheries of [Spec-FocP] and subsequently to [Spec-CP]. When the lexical wh-adjunct remains within v*P, its question features covertly move to [Spec-CP], because the head Foc does not inherit an edge feature from C. In wh-subject extraction, the wh-subject overly undergoes movement to [Spec-CP] because C obligatorily inherits the edge feature to the head Top, which triggers movement of the illogical subject in unaccusative/ergative structures and the logical external specifier in unergative structures to [Spec-CP]. Moreover, Mehri obeys the Phase Impenetrability Condition of Chomsky, where wh-subject and wh-adjunct phrases must pass through certain phases until [Spec-CP].

## Introduction

Mehri is a minority-endangered language spoken in Eastern Yemen by more than 250,000 speakers, particularly in Al-Mahrah region. It is also spoken in southern Oman (Dhofar), parts of central-southern and eastern Saudi Arabia, and in diasporic communities in parts of the

University under the General Research Funding Program Grant Code (NU/DRP/ SEHRC/12/3). The funder had no role in the study design, data collection and analysis, the decision to publish, or the preparation of the manuscript.

**Competing interests:** The authors have declared that no competing interests exist.

Arab Gulf and eastern Africa [1–6]. As a sub-group of the Semitic language family (which is a branch of Afroasiatic family), Mehri is the most spoken of the Modern South Arabian languages (MSALs) in the present Arabian Peninsula. In other words, Mehri is a Southern Arabic language. Before the advent of Islam, Mehri and its sister MSALs were spoken largely in the southern part of the Arabian Peninsula. Moreover, traditional and modern Arabic scholars, who worked on Arabic dialects, stated that Mehri is not only a branch of the Southern Eastern Arabic language spoken in Al-Mahra region and Socotra in Yemen, but also an old Arabic tongue. In Yemen, the Mehri language does not have a formal script and is threatened by the majority language, Arabic. It is not used in government institutions/bodies, as it is not understood by them. The native language of the majority in Yemen is Arabic, not Mehri; the latter is only spoken, but not written, by Mehri people in Yemen and other parts of the Arab Gulf and Africa [2, 6]. Increasingly, endangered language communities are fighting for linguistic justice. In this context, Oberly et al. [6] stated that the indigenous community-member activism for linguistic sovereignty can be noticed in the growth of language revitalization efforts by the Maori [7], Hawaiian [8], and Blackfeet [9]. The objective of this paper is to revitalize and study wh-constructions in the Mehri language in order to provide a modern syntactic analysis of the wh-question movement within Chomsky's [1, 10, 11] Phase-based framework. The language under study is morphologically rich and intrinsically bound to the cultural environment. However, the data collected from Mehri native speakers are authentic and reflect the way of life in Al-Mahrah region in Yemen. Mehri community members have traditionally been involved in agriculture, fishing, and livestock husbandry; Mehri is their native language and is still very much spoken by all of them in their daily life situations. They also speak Arabic when they communicate with other Yemenis and government officials, and they do the same when they meet other Arabs in other places/states. Furthermore, what is fascinating in the Mehri language is that it is an agglutinative language, which has rich morphological features marked overtly on verbs, nouns, adjectives, and modifiers. It appears to exhibit two alternative word orders: SVO and VSO [12]. The marked syntactic orders are broadly used in daily life conversations.

Moreover, some studies have been conducted on Mehri language which investigated the clause structure, morphology and grammar [5, 13, 14] Some other analyses described the syntactic question formation and how wh-words can used in Mehri [13, 15, 16]. Furthermore, there are a few recent studies that examined wh-interrogatives in Mehri within the earlier assumptions of the minimalist framework [4, 17].

Let us first introduce the relevant studies on Mehri and explore what has been done so far. In the early twentieth century, linguists and scholars started exploring Mehri language. This has been seen in the work by [18]. A subsequent work with an impressive grammatical study of Mehri followed later in (1953) by [19] in his syntactic study of Mehri. Furthermore, [16, 20] provided a rich contribution which was considered to be a landmark in the analysis of Mehri language; one of which is the first dictionary of Mehri published in (1987), in addition to articles that focused on Mehri. Two Western scholars, Simeone-Senelle and Lonnet [21] contributed immensely to the analysis of Mehri and Southern Arabian languages during their Austrian South Arabian Expedition. Furthermore, Hofstede [22] worked in his PhD thesis on the syntax of Shehri/Jibbali (a counterpart language of Mehri) at the University of Manchester.

Moreover, the first development in Mehri analysis can be seen in the publication of the first dictionary of Mehri in (1987) by [20]. The second development is also observed in the Grammar sketches written for Mehri in works by [15, 23–25]. Though these published works were useful, they did provide a detailed analysis of syntactic phenomena in Mehri syntax, especially wh-interrogatives; the focus was on phonetic, phonological, morphological, and descriptions. Moreover, in a recent PhD work, that has been done on Mehri is by [14], focused on the

morphology of the Qishn dialect of Mehri, a recent welcome addition to the morphological analysis of Mehri.

Recently, two noteworthy volumes of Mehri were presented by [5, 13], which are useful and helpful in the analysis of Mehri structures. Rubin [13] wrote a comprehensive book titled "The Mehri Language of Oman", where he provided an in-depth grammatical description of the phonology and morphology of Omani Mehri. However, Rubin gave little attention to the syntactic properties of Mehri wh-interrogatives. The other volume is seen in [5] in which she offered recent dual-grammar that aims to contrast the Yemeni and Omani dialects of Mehri, and discussed their phonology and morphology in addition to grammatical aspects of their syntactic structure. However, Watson did not describe question formation in Mehri, and wh-movement was left undiscussed in the second volume.

Furthermore, Alrowsa [4] presented a grammatical description of Mehri wh-interrogatives based on [4, 13, 15, 16] provided a syntactic description of wh-phrases in argument and adjunct positions and illustrated the differences between these syntactic positions. His study concluded with an analysis of yes-no question formation in Mehri. In addition, Alrowsa [4] mentioned that there are two types of wh-phrases in Mehri: wh-phrases that can appear alone, as in (1), and those that can be used as modifiers of another noun phrase, as in (2).

1a. hæh/hæʃən 'what'
 b. wkuh 'why'
 c. hibuh 'how'
 d. ħuh 'where'
 e. kɛm 'how much/many'
 f. mɔn 'who'
 g. majtən 'when' [4, p. 126]
2a. hæh/hæʃən ṭəbx-ut fəṭməh
 what cook.pf-3f.sg Fatimah
 'What did Fatimah cook?'
 b. hæh/hæʃən ʃin-ək
 what see.pf-2m.sg
 'What did you see?' [4, p. 127]

Alrowsa [4] stressed that Mehri permits optional wh-word movement. However, Alrowsa [4] assumed that Mehri wh-phrases in the argument and adjunct positions can be fronted to the clause left periphery or left in situ. Alrowsa's [4] analysis is brief and descriptive in nature and but it does not offer an in-depth explanation of wh-interrogatives and how wh-movement is derived in the minimalist syntax.

In a recent study, Al-Qumairi [17] examined wh-questions in transitive and ditransitive structures in Mehri. Al-Qumairi [17] mentioned that Mehri permits two kinds of wh-questions: the argument wh-questions and the adjunct wh-questions. In addition, Al-Qumairi [17] stated that two wh-strategies can be observed in Mehri syntax: wh-fronting to the clause left periphery and wh-in-situ.

Moreover, to the best knowledge of the researchers that none of the prevoius studies examined the syntax of wh-questions in (un)accusative and (un)ergative structures in Mehri whithin the framework of Chomsky's Phase Theory. Most of the previous studies were descriptive in nature and did not provide an adquate explanation of wh-questions in Mehri. None of the precdeing analyses explored how Mehri wh-phrases undergo movement within the Phase Theory approach. Therefore, the current paper attempts to offer a unified analysis of Mehri wh-questions in the structural projections of unaccusative/ergative and unergative verbs. Unaccusative/ergative and unergative verbs are one-place predicates. They are traditionally called intransitive verbs. Unaccusative/ergative verbs in the Mehri language are derivatives, in

which the external argument is not a true subject. They are derived from accusative verbs. The affixed element -**ta**- (attached to verbs) is the unaccusative marker. This can be exemplified in (3).

3. Accusative
Two-place predicate
fiṣ̌ṣ̌
ḥiṣṣ
riṣ̌ṣ̌
ḳiṣ̌ṣ̌
raḥāṣ
ḥamōṣ
Unaccusative
One-place predicate
fataṣ̌
ḥataṣ
rataṣ̌
ḳataṣ̌
rataḥaṣ̌
ḥatamaṣ̌
Gloss
explode
roughly tie
partly break
damage
wash
sour

Contrastively, the unergative verbs select an external true subject that theta-marks an agent or experiencer theta-role. This can be demonstrated in (4).

4. Unergative
One-place predicate
šaḥrōd
sayūr
ḏalūf
baḳūṣ̌
šawkōf
śaxawlōl
Gloss
become tired
go
jump
run
sleep
set down

It attempts to demonstrate whether or not the Mehri language permits overt wh-movement in its syntactic description. This study also seeks to examine the interaction between Mehri wh-questions and Chomsky's [1, 10, 11] Phase-based framework. The topic of Mehri wh-questions has been selected for study for the following reasons: (i) The syntax of Mehri wh-questions has not been examined yet; (ii) To explain how wh-questions in Mehri unaccusative/ergative and (un)ergative structures can be accounted for within Chomsky's [10] Phase-based

framework; (iii) There is a morpho-syntactic need to provide a modern analysis of wh-questions in Mehri in order to show how the interrogative structures can be derived, how wh-phrase movement can be established, and what motivates Mehri wh-phrases to move from their canonical positions to [Spec-CP]. Therefore, this work seeks to answer the following research questions within Chomsky's [1, 10] Phase-based Theory: (i) Does Mehri language allow fronting of wh-phrases to [Spec-CP]?, and (ii) how can wh-question movement in Mehri unaccusative and (un)ergative structures be accounted for?

Moreover, this paper follows the following organizational structure: The introduction section provides an overview of the study. The literature review section surveys the previous analysis of wh-constructions in Mehri. It also discusses the syntactic approaches to the analysis of unaccusative/ergative and unergative structures in Arabic and other languages. The theoretical framework section introduces Chomsky's Phase Theory [10, 11]. The analysis section scrutinizes wh-questions in the Mehri language and investigates the extraction of wh-phrases from unaccusative/ergative and unergative structures. Finally, the conclusion section wraps up the study.

## Literature review

### Previous work on Mehri

Alrowsa [4] discussed question formation in Mehri, a Modern South Arabian Language, where he demonstrated that Mehri exhibits optional wh-fronting in which a wh-question word may appear in situ or undergo movement to the front of a clause. Alrowsa [4] indicated that that Mehri has the so-called optional wh-movement, which is similar to what has been discussed in other Semitic languages such as Egyptian and Lebanese Arabic. Alrowsa [4] argued that "Mehri does not exhibit wh-movement per se, but that fronted wh-words in the language are based generated in fronted positions as part of clefting and topicalization structures, and are licensed via unselective binding and not by wh-movement" (p. 18). In this study, Alrowsa [4] adopted the set of categories of analysis discussed in languages with Basic Linguistic Theory by [26], building on descriptions of other languages in the Semitic family. In the final chapter of his PhD thesis, Alrowsa [4] briefly based his analysis on the earlier framework of the Minimalist approach in the discussion of optional wh-fronting in Mehri.

Furthermore, Alrowsa [4] indicated that Mehri allows optional wh-question movement. However, wh-phrases appear to stay in situ when an argument/or adjunct is being questioned; here, the wh-phrase and intonation illustrate that the clause is a wh-interrogative. On the other hand, Alrowsa [4] stressed that the wh-word can also be fronted to the clause-initial position. This can be demonstrated in (5).

5a. hət tə-ɣɔrəb mɔn
you.m.sg imp.2m-know who
'Whom do you know?'
b. mɔnə hət tə-ɣɔrəb
who you.m.sg imp.2m-know
'Whom do you know?' [4, p. 128]

Alrowsa [4] described types of wh-questions in the subject and object argument positions. Wh-subject extraction cab be illustrated in (6), where the subject wh-word is fronted to the clause-initial position.

6a. ṭəbx-ɔt mɔn æ-qut
cook.pf-3f.sg who def-food
'Who cooked the food?'
b. mɔn ṭəbx-ɔt æ-qut

who cook.pf-3f.sg def-food

'Who cooked the food?' [4, p. 132]

On the other hand, wh-object questions can be demonstrated in (7), where no movement to the front of the clause-initial position is necessary.

7a. fəṭməh ṭəbx-ut hæʃən

Fatimah cook.pf-3f.sg what

'What did Fatima cook?'

c. ṭəbx-ut fəṭməh hæʃən

cook.pf-3f.sg Fatimah what

'What did Fatima cook?' [4, p.134]

However, Alrowsa [4] indicated the object wh-phrase can also be fronted clause-initially, as illustrated in (8).

8a. hæʃən ṭəbx-ut fəṭməh

what cook.pf-3f.sg Fatimah

'What did Fatima cook?'

b. hæʃən fəṭməh ṭəbx-ut

What Fatimah cook.pf-3f.sg

'What did Fatima cook?' [4, p. 134]

In his analysis of wh-fronting in Mehri, Alrowsa [4] mentioned that wh-words in the argument and adjunct positions in Mehri can be fronted or left in situ. Although Alrowsa's [4] analysis was briefly built on the earlier assumptions of the minimalist framework, he did not provide an in-depth account of wh-movement in Mehri as his analysis was basically descriptive in nature.

In answering the question of whether or not fronting of question words in Mehri is an instance of wh-movement or not, Alrowsa [4] argued that wh-subject and object interrogative structures "do not involve movement, but rather license interrogatives via unselective binding in the sense of [27, p.159]. Alrowsa [4] pointed out that Mehri wh-questions are insensitive to islands and assumed that wh-movement is not involved. Moreover, following Soltan's [28] analysis of optional wh-movement in Egyptian Arabic, Alrowsa [4] stressed that, like Egyptian Arabic, "Mehri does not license interrogatives via movement, but rather than unselective binding" (p.161). Alrowsa [4] argued against fronting wh-movement in Mehri and stressed that "fronted wh arguments are not derived via wh-movement, but rather by (sometimes hidden) cleft constructions" (p, (p.161). Alrowsa's [4] argument follows Soltan's [28] analysis of Egyptian Arabic. However, Alrowsa [4] mentioned that the fronting of wh-adjunct questions is not done via clefting, but rather via topicalization. This means that Mehri is different from Egyptian Arabic, in the latter (Egyptian) wh-fronting/topicalization of adjuncts is very much marginal.

In a recent study on Mehri, Al-Qumairi [17] sought to describe the syntactic and morphological properties of Mehri, where he adopted the Minimalist framework in examining the morpho-syntactic properties of the grammatical categories and exploring the left periphery and movement of the clause structure in Mehri. Moreover, Al-Qumairi [17] examined movement and the left peripheries in Mehri clausal constructions. In the final section of his study, Al-Qumairi [17] discussed wh-interrogatives in transitive and ditransitive structures in Mehri within the Minimalist approach, where he indicated that there are two types of wh-questions: the argument wh-questions (that have external and internal arguments) and the adjunct wh-questions. Al-Qumairi [17] observed that there are wh-strategies in Mehri wh-movement: wh-fronting and wh-in-situ.

## Previous analysis of wh-constructions in Arabic

This section reviews the previous analyses conducted by Arab scholars on wh-questions in Arabic syntax. The reasons why this section begins reviewing the previous analysis of wh-questions in Arabic can be attributed to the following: (i) both Mehri and Arabic belong to the Semitic language family, which also includes Aramaic, Amharic, Syriac, Canaanite, and southern Arabian languages- one of which is Mehri [13]. (ii) Both Mehri and Arabic exhibit two prominent word orders: SVO and VSO. Apart from Arabic that exhibits agreement asymmetry between a verb and its post-verbal subject within VSO order, Mehri appears with a full agreement in both SVO and VSO word orders [12]. (iii) Both Mehri and Arabic exhibit certain syntactic properties with regard to wh-movement in unaccusative and (un)ergative structures. The analysis of Arabic wh-questions reviewed in this subsection will facilitate the understanding of the syntactic discussion of wh-questions in Mehri's unaccusative and (un)ergative structures. Moreover, Arabic and Mehri indeed are related languages, but it does not mean that the syntactic assumptions in the present study about Mehri will be based on what has been established in Arabic. What distinguishes the analysis adopted in this study from those on Arabic wh-questions reviewed below is that the analysis about Mehri is based on the Phase Theory of [10], whereas the previous accounts on Arabic wh-question discussed below adopted the Government and Binding theory and early minimalist frameworks. Consequently, the approaches of the analyses are different and the findings are dissimilar.

Based on [1, 29] Minimalist analysis of wh-movement, [30–34] emphasized that wh-movement in Standard Arabic is permitted only with VSO order. [33, 34] stressed that due to feature checking on the left periphery, the subject/object wh-phrase undergoes an obligatory overt movement to [Spec-CP]. Assuming Chomsky's [29] minimalist account of feature checking, Fakih [31] proposed that "raising of a wh-operator to [Spec-CP] is driven by the need for a morphological Q-feature to be licensed" (p.146). This means that a given wh-phrase is triggered to move to the left periphery in order to check an affixal Q-feature realized on the abstract C head position. On the contrary, this assumption may not properly be useful to analyze the wh-in situ questions employed in some Arabic dialects such as Jordanian Arabic [35], Palestinian Arabic [36], and Egyptian Arabic [37].

Furthermore, Alotaibi [38] assumed that "the preverbal NP in SVO is base-generated as a topic, rather than a subject. The moved wh-non-subject moves to [Spec-FocP], while the wh-subject is base-generated in [Spec-TopP]" (p.1). His assumption can be exemplified in the following structures in (9).

9 Ahmad-u 'akala t-tuffahat-a
Ahmad-Nom. ate.3sm the-apple-Acc.
'Ahmad ate the apple.'
10 *maathaa ahmad-u 'akala?
What Ahmad-Nom. ate.3sm
11 *'akala Ahmad-u t-tuffahat-a*
ate.3sm Ahmad-Nom. the-apple-Acc.
'Ahmad ate the apple.'
12 maathaa 'akala ahmad-u?
What ate.3sm Ahmad-Nom.
'What did Ahmed eat?' [38, p.13]

In (9), if we assume that *Ahmad-u* is a preverbal subject (termed *fā'il muqaddam* in Arabic), it may then move to [Spec-TopP]. Rather, the wh-phrase *maathaa* 'what' is not allowed to move to [Spec-CP], as in (10). This, however, is allowed when the verb *'akala* 'ate' undergoes

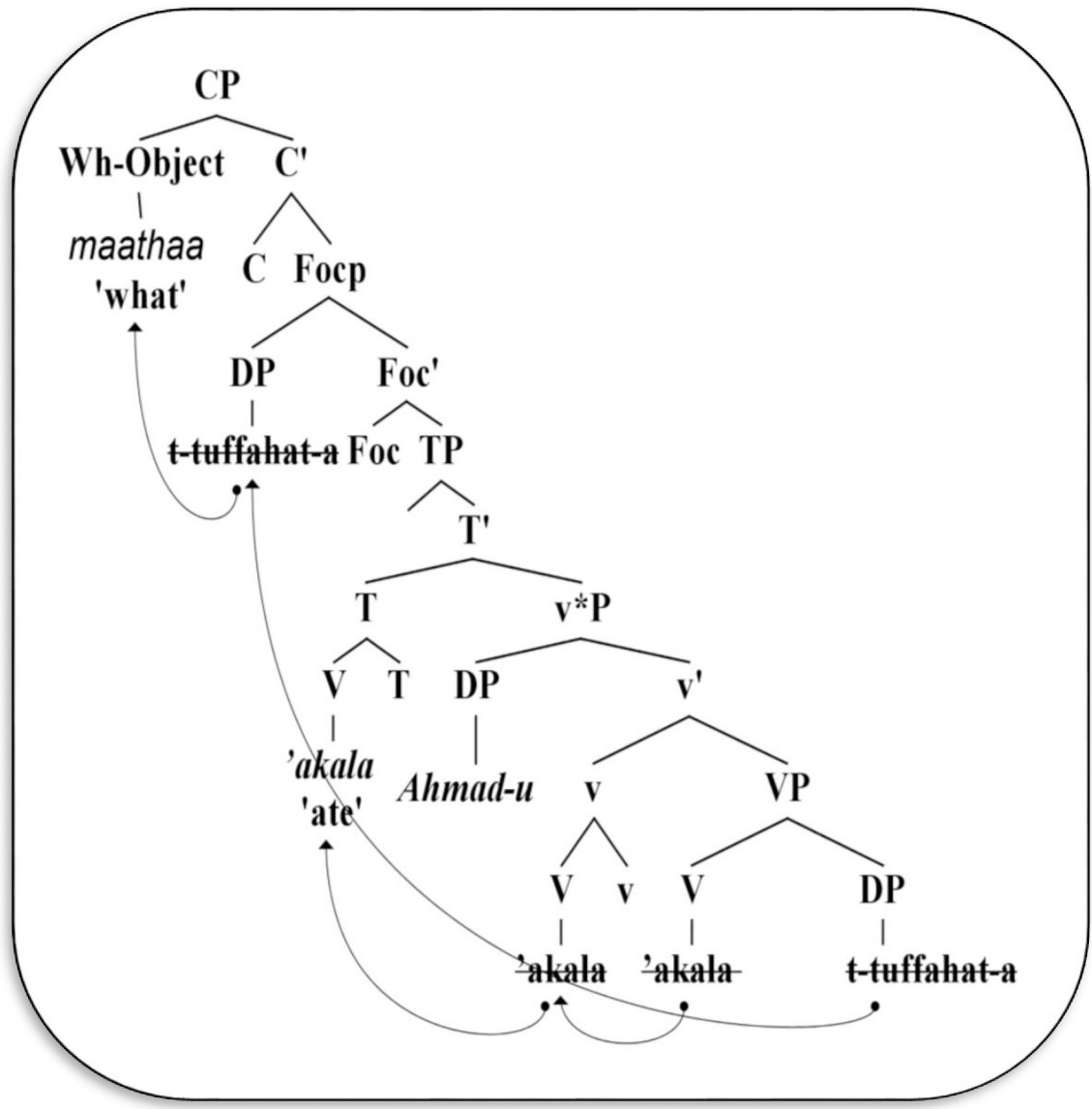

**Fig 1. Derivation of wh-non subject phrase in Arabic.**

movement to T, as in (11), while [Spec-TP] is null and the subject remains in situ within v*P, as in (12). This can be simplified in Fig 1 below.

The object phrase *t-tuffahat-a* 'the apple' is triggered to move to [Spec-FocP] in VSO, while the subject *Ahmad-u* remains in situ. It occupies the specifier position within the v*P phase, whereas [Spec-TP] is null. This movement yields an OVS structure such as (13).

13 t-tuffahat-a, 'akala ahmad-u

the apple, ate Ahmed

'The apple, Ahmed ate.'

According to Aoun [36, p. 206], this type of structure is called a "fronted focus phrase". It is a dislocated structure, not a default. Furthermore, the preverbal object is replaced by a wh-object *maathaa* 'what', and then moves to [Spec-CP], as illustrated in (12) above.

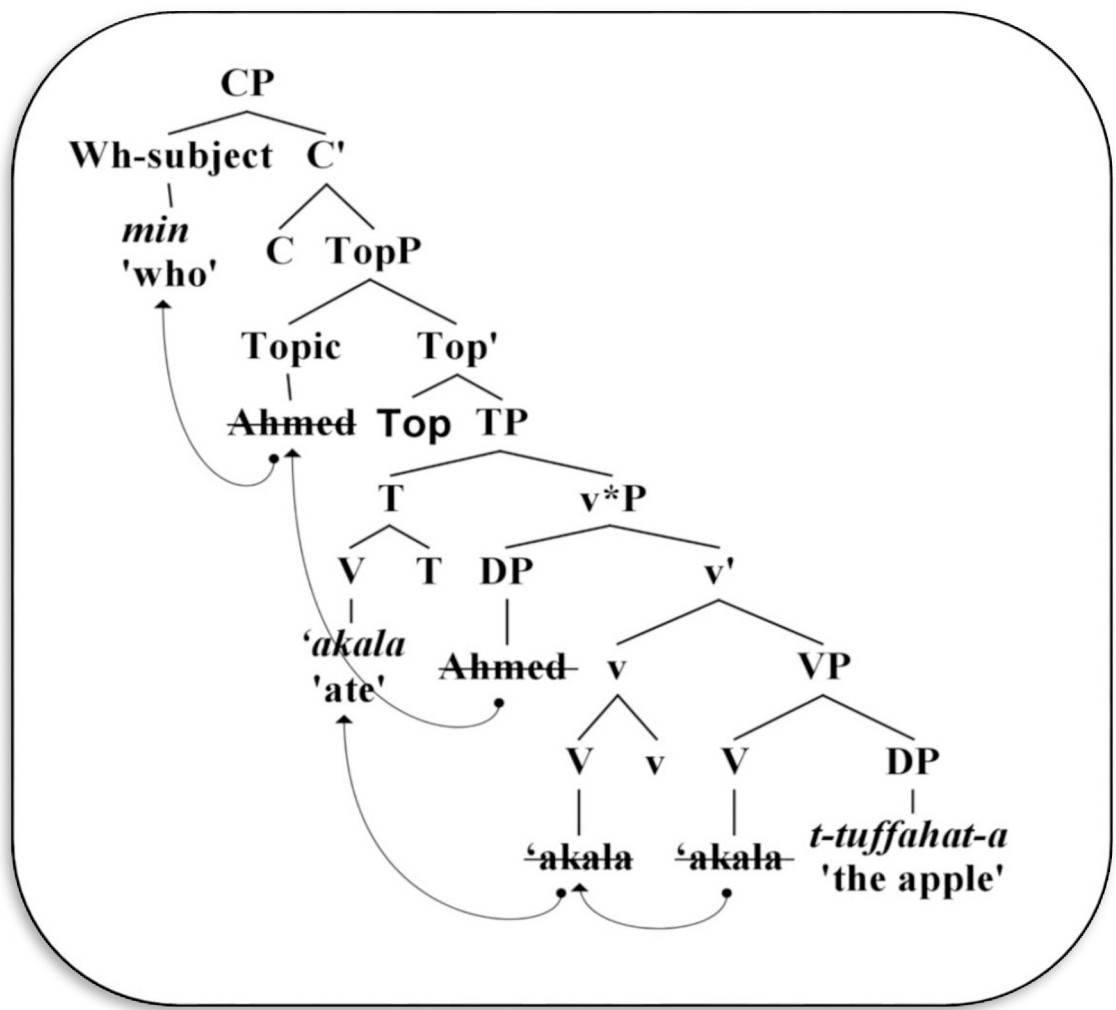

**Fig 2. Derivation of wh-subject phrase in Arabic.**

Alotaibi [38] also indicated that "a problem arises when the non-subject wh-phrases move over the SV order" (p.1). Therefore, the preverbal DP in SVO order is considered a Topic (*mubtada'*), while the post-verbal DP is merely a subject (*fā'il*) located within the v*P phase. Specifically, [Spec-v*P] will be topicalized and then will move to [Spec-TopP], thus forming a TopP in which the preverbal DP is a Topic rather than a subject [35]. This topic has to be replaced by a wh-subject such as *min* 'who', and then moves to [Spec-CP], thereby producing a wh-subject question, as demonstrated in (14), and syntactically diagrammed in Fig 2 below.

14 min 'akala t-tuffahat-a

who ate.3sm the-apple-Acc,

'Who ate the apple?'

Furthermore, in his analysis of wh-interrogatives in Standard Arabic, Al-Shorafat [39] provided an investigation into the syntax of wh-questions; he assumed that T in VSO does not project an edge feature that is solely responsible for attracting [Spec-v*P]. This [Spec-v*P] remains within the v*P phase and inherits its agreement features to [Spec-TP], that is, *pro*. [35], on the other hand, postulated that C is the source of morphological features such as Q, Agree, Case, Topic, and Focus. C inherits its features to all functional heads (i.e., T, Foc, and

Top) in its domain. Similarly, Fakih [19, 33] was in agreement with Musabhien's [35] analysis and, in turn, stated that C is an indispensable category that determines the force of the sentence and its syntactic structure, namely, it inherits an edge feature to all prior heads, and triggers movement to the local goals. As a result, a declarative clause can be constructed with either SVO or VSO, whereas an interrogative clause appears with a wh-specifier that includes an invisible case feature. With regard to the occurrence or non-occurrence of the edge feature on C, wh-phrases have to overtly or covertly move to the left periphery of the interrogative clauses. This results in two types of wh-interrogatives: wh-fronting questions and wh-in situ questions.

Furthermore, Arab linguists such as Alonini [40] indicated that intransitive verbs in Standard Arabic follow the same classification found in English unergative and unaccusative verbs. Alonini [40] observed that the sole argument of an unaccusative verb gets its nominative case in an in-situ position; she attributed this to Locality of Matching, where morpho-syntactic agreement must exist between nominative case on the head T and the subject NP in the internal argument position of VP projection. The nominative case assignment follows from the fact that there is no intervening NP between T and the sole argument in the internal subject position of VP, which is a base-object position [40].

Alrashed [41] discussed unaccusative verbs and indicated that the Arabic unaccusative verb does not have an object or an external argument. Alrashed [41] viewed its sole argument as only a subject NP positioned in [Spec-VP] configuration. Furthermore, in his examination of causation alternation verbs in Arabic, Al-Qadi [42] assumed a different analysis of the little v (erb) in order to express a different event. Al-Qadi [42] suggested a voice head in order to convert the root to a verb or noun; the voice head is located between the TP projection and the vP phase. Moreover, Ben Ayeche [43] explored the passive verb in Arabic from a Chomskyan minimalist perspective; he observed that *kusira* 'was broken' was derived with an unvalued voice feature, which is later valued by the 'VoiceP' that has a passive morpheme (u-i). Ben Ayeche [43] noted that the VoiceP is a weak phase and lacks an agent theta role. In the absence of an intervening NP between the subject NP and the head T, the latter (T) assigns a nominative case to the sole subject NP via the operation Agree; nominative case assignment is done in-situ [43]. Furthermore, Abdel Wahed [44] analyzed nine lateral verb forms in Standard Arabic and noted that three forms showed the features of an unaccusative verb, and in these three forms, a sole argument is located in the complement position of the v*P phase, which bears a thematic role patient/theme and moves to [Spec-vP] position to receive nominative case, being c-commanded by T.

## Syntactic approaches to the analysis of unaccusative and (un)ergative) structures

The syntactic analysis of unaccusative/ergative and unergatives in the world languages has received considerable attention in the past decades in generative grammar. Let us discuss the main approaches that dealt with unaccusative/ergative and unergatives in the past four decades. The first approach began with Perlmutter's [45] seminal work on unaccusatives. The origin of 'unaccusativity' started in Perlmutter's [45] Unaccusativity Hypothesis (UH); where he classified intransitive verbs into two sub-classes: unaccusatives and ergatives, which were viewed as two syntactically distinct classes of intransitive verbs. It was assumed that unaccusative verbs have an underlying object that moves to the subject position, while unergative verbs have based-generated subjects in the surface structure [45, 46]. Perlmutter [45] assumed that the mapping of the sole argument onto syntax as a subject or direct object can semantically be predictable. Based on the HU, a semantic and syntactic distinction had been made between

unaccusative and ergative verbs. Moreover, an intransitive verb is viewed as a one-place predicate that has one argument [47]; this argument bears one (patient/theme) theta role for unaccusative verbs and an agent theta role for unergative verbs [48].

The second approach stemmed from Chomsky's Government and Binding (GB) framework. In this GB approach, Burzio [46] made a distinction between two intransitive verbs via verbs' theta-marking features. Burzio [46] assumed that the sole argument in unaccusative verbs is the same as the D-structure object, while in unergative verbs it is the same as the agent at the s-structure, as illustrated in (15).

15a. Unergatives: NP [$_{VP}$ V] Tony danced.

b. Unaccusatives: [$_{VP}$ V NP] Tony arrived.

In unergatives, the subject NP (Tony) carries an *agent* theta role, as in (15a), while in unaccusatives, the subject (Tony) carries a *theme* theta role, as in (15b). Unergatives and unaccusatives have different underlying syntactic structures. For instance, in unergatives, the VP has an external argument NP and does not need a direct internal argument. On the contrary, in unaccusatives, the VP has an internal argument NP, but not an external argument. The inability of the verb to assign an agent theta role to the subject NP can be attributed to the lack of an external argument, and, hence, there is no possibility of assigning an accusative case [46]. The only candidate to be assigned as a nominative case is the sole argument by the intransitive verb of the clause.

The third approach was proposed by [49], who assumed that the difference between unaccusatives and unergatives is syntactically realized in the deep structure representations. It can also be semantically decided in the syntax. Levin et al. [49] indicated that any framework that posits a specific syntactic distinction between unaccusatives and unergatives might have to stress that each verb would have two argument structures: one projecting the unaccusative deep structure representation, while the other one projecting the deep structure configuration for unergatives. Levin et al. [49] formulated a lexical argument-structure changing rule responsible for deriving unaccusatives from the unergative manner of motion verbs during syntactic analysis.

## Phase-based framework

This section introduces Chomsky's [1, 10, 11] Phase-based approach. In this minimalist framework, a given clause (declarative or interrogative) is assumed to proceed through two successive phases: inner v*P phase and outer CP phase, as represented in the following standard structure:

From Fig 3, the syntactic representation contains two phases: the inner v*P phase and the outer CP phase. The outer CP phase inherently shows the type of clause (whether it is declarative, interrogative, imperative, or exclamative), while the inner v*P phase locates a complementary position that undergoes transfer to CP. Based on Chomsky's phase-based analysis and in line with [33, 39, 51] the clause structure has been divided into two propositional phases: clausal CP and transitive v*P. The clausal CP phase represents a complete syntactic projection including force interpretation (declarative, interrogative, etc.), whereas the v*P phase represents a complete argument/thematic structure of verbs including a logical real external argument, as the opposite of passive and ergative verbs, which are defective to form a v*P phase. The minimalist analyses maintained that C and v categories are functional heads of their projections, and the syntactic operation involves an agreement between these heads and their domains. This argument is in line with Chomsky [1]'s assumption, which states that the "core functional categories C, T and v" are active probes, which contain uninterpretable features that require valuation via the Agree operation." (p.102). Given this, Chomsky [10]

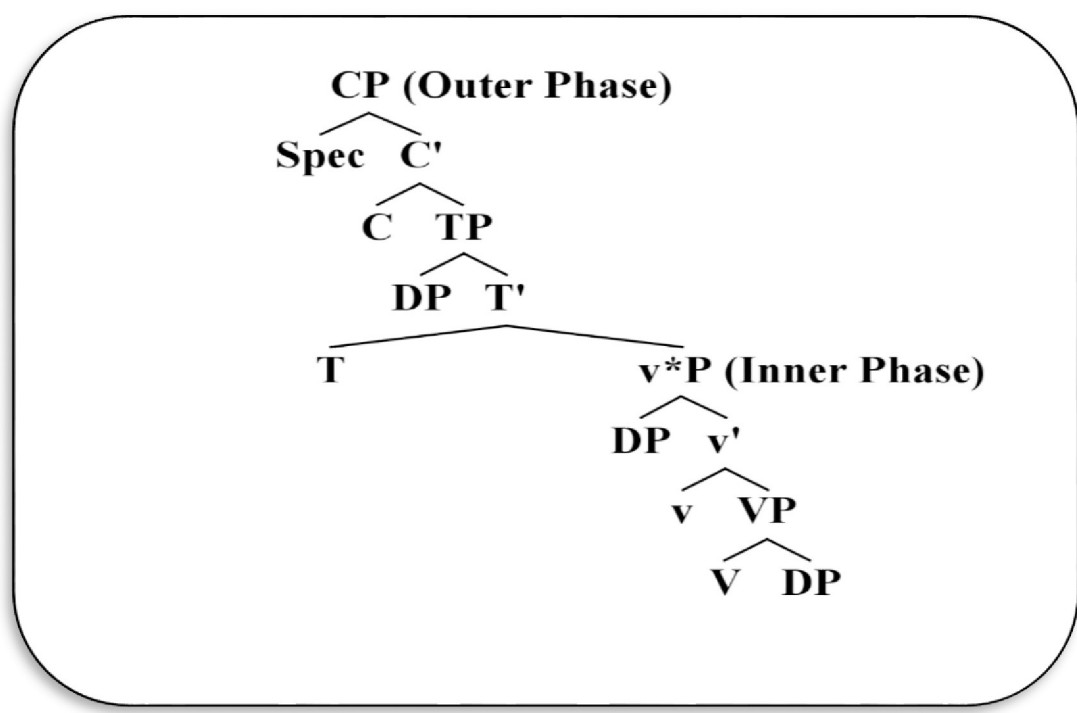

**Fig 3. Chomsky's Phase-based framework (adopted from Chomsky, [50]).**

presumed a "transmission of the Agree feature" (p.148), where he argued that the functional heads C and light v should transmit their Agree features to the heads T and V while deriving the structure. Hence, CP and v*P phases are probably formed, giving a sense of the outcome structure.

As for the projections TP and VP, Chomsky [1, 10] indicated that these projections are not phases because they are in complementary domains and are contained within CP and v*P. Chomsky indicated that all complements (even v*P phase) undergo transfer simultaneously to PF and LF components in the CP domain. Given this, Chomsky [1] stated that in the Phase Impenetrability Condition "in phase *a* with head *H*, the domain of *H* is not accessible to operations outside *a*, only *H* and its edge are accessible to such operations" (p.108).

## Rizzi's [52] Split-CP hypothesis

In this sub-section, we introduce Rizzi's [52] seminal analysis of the fine structure of CP. The objective of introducing Rizzi's Split-CP is that parts of the syntactic analysis of the present study adopted Rizzi's proposal, especially TopP and FocP projections. A closer look at the traditional understanding of the clause reveals that it consists of three syntactic layers: CP, IP, and VP. As the CP indicates a speaker's attitude, the IP illustrates grammatical input (tense and agreement features). The VP, on the other hand, contains arguments. Each of these layers has been split into separate projections; the CP is split in [52], the IP is in [53], and the VP in [54]. In the last two decades, Rizzi [52, 55], and Al-Shorafat [39], to name a few, proposed a syntactic expansion of a functional projection in order to accommodate the complementizer (C) and other syntactically relevant material existing on the clause left periphery. Rizzi [52] suggested that the CP projection of a clause structure should be split into a number of separate functional projections in the syntax in order to host elements that surface on the clause's left

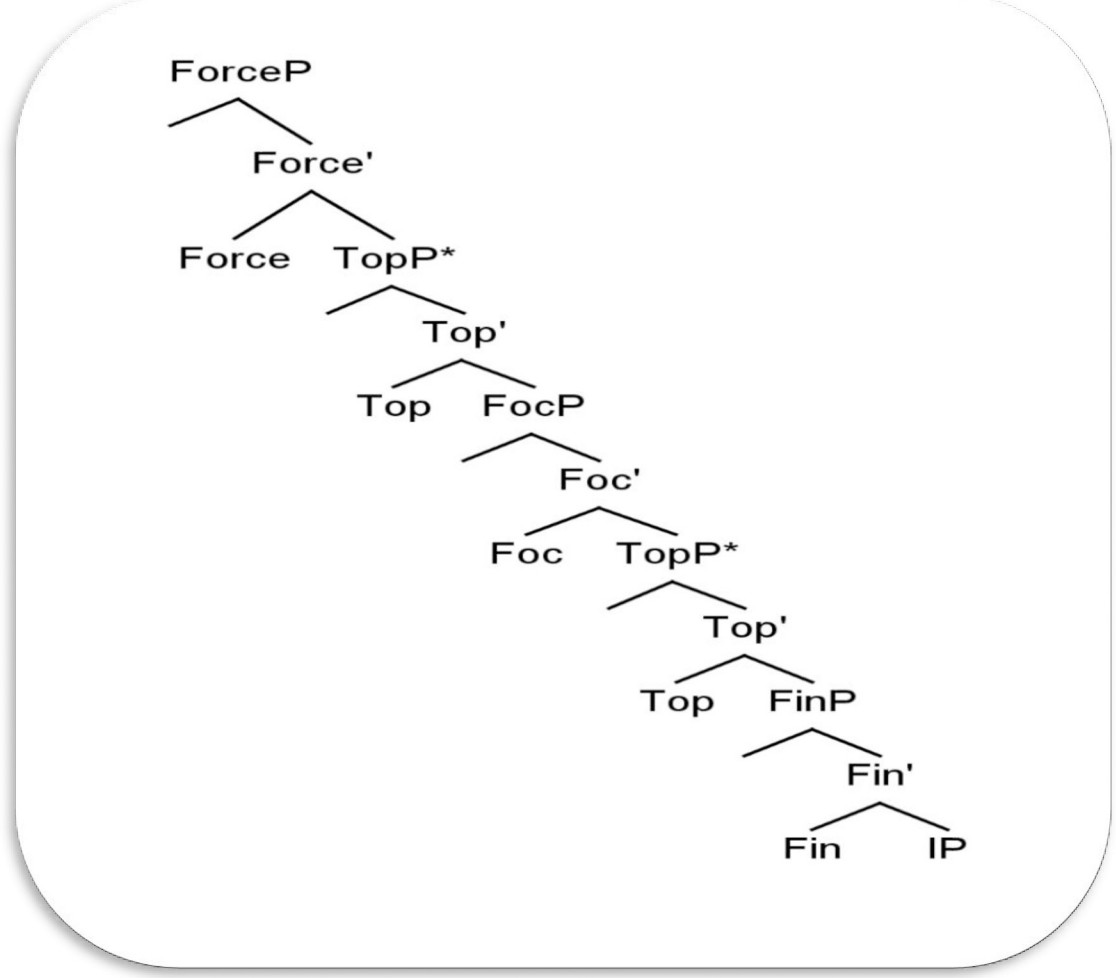

**Fig 4. Tree structure of Rizzi's [51] Split-CP hypothesis (adopted from Rizzi, 1997, p. 297).**

periphery. In Rizzi's [52] analysis, a head complementizer (C) was viewed as a Force marker, thus heading Force Phrases. Furthermore, topicalized and focused constituents were viewed as different Topic Phrases (headed by Top) and Focus Phrases (headed by Foc). Hence, the outcome of the proposal is that there are separate projections, each one having a specifier position and a head on clause structure. Rizzi's Split-CP structure can be illustrated below in Fig 4.

A closer look at Rizzi's [52] hierarchical ordering of the expanded functional projections reveals that the projection ForceP dominates TopP, which dominates FocP, which dominates FinP; this would look like (. . . Force . . . (TopP) . . . (Foc) . . . FinP). Moreover, Rizzi [55] modified the preceding proposal, where he added IntP (interrogative projection) to the clause structure to account for wh-questions. It has been demonstrated in the existing literature that in Germanic languages topics undergo V-movement, whereas they do not in English; in the latter (English), the head of the Topic does not need to be filled in the syntax.

### Analysis of wh-questions in (un)accusative/ergative structures in Mehri

Compared to other endangered languages, not much attention has been paid to the syntax of wh-questions in Mehri. This section seeks to examine wh-questions in (un)accusatives/

ergative structures in Mehri. This study aims to provide an adequate answer to the following research questions: (i) Does Mehri language allow fronting of wh-phrases to [Spec-CP]? and (ii) how can wh-question movement in Mehri unaccusative and (un)ergative structures be accounted for neatly within Chomsky's [1, 10] Phase-based Theory?

In their syntactic analysis of Mehri, Rubin [13] and Watson [5] indicated that verbs in Mehri are deemed complicated; their radical roots are either to be trilateral; √ṭbr 'to break', quadrilateral; √dmdm 'to grope', or quinquiliteral; √fǵnẓẓ 'to start crying'. In Semitic languages, the verb roots are indicated by a sequence of consonants (dubbed as radicals); this invites the terminology 'consonantal root.' In Arabic, for instance, consonantal roots are used to form words by the addition of vowels and non-root consonants. There also are other terms used in Arabic such as, "trilateral roots" and quadrilateral roots". A trilateral root (dubbed as 'triconsonantal root') is viewed by Arab and Western scholars as a root that has a sequence of three consonants (e.g., k-t-b ك-ت-ب "to write"). A quadrilateral root, on the other hand, is a consonantal root that has a sequence of four consonants. A quadriliteral form is a word derived from four consonantal roots in Arabic (e.g, the quadriliteral root عرقل 'to obstruct'). This property is a Proto-Semitic feature that distinguishes all Semitic languages from other languages. Traditionally, Rubin [13], Simeone-Senelle [56], and Watson [5] classified Mehri verbs into transitive and intransitive types. As far as we know, Mehri's unaccusative/ergative and unergatives structures have not been analyzed within modern generative syntax. This study seeks to explore the syntax of wh-questions in Mehri. In this paper, however, Mehri wh-phrases will be restricted to those extracted from unaccusative/ergative and unergative structures.

Let us explore ergativity in Mehri. Generally, ergativity "refers to a system of marking grammatical relations in which intransitive subjects pattern together with transitive objects" [57, p. Intro.]. It is known that intransitives are verbs that do not assign internal objects. Carnie [58] indicated that the unaccusative/ ergative verbs are a subtype of intransitives that assigns only one theta role to the selected argument NP, which does not bear accusative case. In other words, ergative verbs assign theme theta role to the extracted subject and check nominative case [59]. The extracted nominative subject is not the real subject of the structure. Rather, it is an illogical subject that makes it behave similarly to the syntactic object of the accusatives or the moved subject of the passive predicate. In Mehri, the unaccusative/ergative verbs are morphologically marked, as illustrated in (16) derived from (17).

16 anīʕāytan ḏa-ḥāyabīt fataṣ̌ṣ̌
[the-breasts of-the-camel-3fp.Nom.] [exploded-fp.Perf.]
'The camel's breasts exploded.'

17 aŝxōf faṣ̌ṣ̌ anīʕāytan ḏa-ḥāyabīt
[the-milk-3ms.Nom.] [explode-3ms.Perf] [the-breasts of-the-camel 3fp.Acc.]
'The milk exploded the camel's breasts.'

(16) shows that Mehri has a rich agreement inflection between the verb and its participants. For example, in (16) both fataṣ̌ṣ̌ 'exploded' and the preverbal DP anīʕāytan ḏa-ḥāyabīt 'the camel's breasts' bear similar agreement features. These features are third person, feminine, and plural. On the other hand, the preverbal DP aŝxōf 'the milk' in (17) agrees with the verb faṣ̌ṣ̌ 'exploded' in third person, masculine and singular features. Derivationally, the verb fataṣ̌ṣ̌ 'exploded' in (16) is an unaccusative/ergative type. The verb fataṣ̌ṣ̌ 'exploded' is overtly marked by an infix [-ta-]. Given the verb morphology in Mehri, we argue that the [-ta-] is an ergative marker. It is added to the accusative verb faṣ̌ṣ̌ 'exploded', and semantically changes to the unaccusatives/ ergative type. While the accusative verb faṣ̌ṣ̌ 'exploded' obligatorily assigns two arguments: subject and object (bearing agent and theme theta-roles, consecutively), the unaccusative/ergative verb fataṣ̌ṣ̌ 'exploded' licenses only one external argument theta-

marked a theme theta-role. It is the DP *anīʕāytan ḏa-ḥāyabīt* 'the camel's breasts.' This extracted argument is not a logical subject of the verb. But rather, it is originally located in the object position of the accusative verb *faṣṣ* 'exploded', as demonstrated in (17).

Moreover, let us distinguish between a causative verb and an (un)accusative verb in Mehri. It can be made clear that the verb *faṣṣ* 'exploded' is not a causative verb, but rather an accusative verb that selects two arguments: the external argument (i.e., subject) and internal argument (i.e., object). The V 'exploded' assigns the external argument *aṡxōf 'the milk'* and internal argument *anīʕāytan ḏa-ḥāyabīt* 'the camel's breasts.' On the other hand, the unaccusative V *fataṣṣ* 'exploded' assigns one argument, it is the external argument *anīʕāytan ḏa-ḥāyabīt* 'the camel's breasts.' In contrast, the causative verbs in Mehri are marked by the suffix *ha-*, as shown in following examples in (18):

18a. aĝayg habhōl ḥïrḗz baśïwūṫ

the-man matured rice with fire

'The man matured the rice with fire.'

ḥïrḗz bahḗl

the-rice matured

'The rice matured.

In (18a) the causative V *habhōl* 'matured' (which is derived from the intransitive V *bahḗl* 'matured' in (18b)) assigns three arguments: the subjective agent *aĝayg* 'the man,' the thematic object *ḥïrḗz* 'the rice,' and the instrumental object *baśïwū*ṫ 'with fire'.

## Wh-phrase extraction from unaccusative/ergative structures

Let us first examine the working mechanism for wh-questions in active constructions and whether it is overt wh-movement to [Spec, CP] or wh-in-situ. A closer look at Mehri wh-interrogatives reveals that it exhibits two mechanisms for wh-questions: overt wh-question movement and wh-in-situ questions. Let us consider the following examples in (19a-c) to illustrate the point.

19a. Ali tïḳ ḥamōh SVO

Ali drink the-water

'Ali drank the water.'

haśan Ali tïḳ?

what Ali drink.

'What did Ali drink?'

Ali tïḳ haśan?

Ali drank what

The examples in (19b) and (19c) demonstrate active constructions in SVO word order, which show wh-questions. In (19b), the wh-object word *haśan* 'what' undergoes overt movement to the left periphery of the clause for feature valuation; it moves to the [Spec-CP] projection, triggered by the strong wh-feature on the head C of CP. In (19c), on the other hand, the wh-object word *haśan* 'what' (which replaces the object NP ḥamōh 'the water') remains in-situ; it does not move to [Spec, CP] in the overt syntax. as illustrated in the following examples in (20).

20a. aĝayg ṯabūr axalfḗt. SVO

the-man broke the-window

'The man broke the window.'

b. aĝayg ṯabūr haśan? Wh-in-situ

the-man broke what

c. haśan aĝayg ṯabūr? Overt wh-movement

what the-man broke
'What did the man break?'

Based on the Mehri SVO examples in (20), the authors observe that it is possible to resort to the wh-in-situ mechanism, as shown in (1b). Moreover, there are no semantic differences between wh-in-situ in (20b) and wh-movement in (20c). In (20b), the wh-object remains in-situ, while it undergoes movement to the left periphery of the clause in (20c). Besides, in (20b), the wh-object lexically stays in-situ, while the interrogative force determines the semantic reading of the clause.

Moreover, concerning the wh-questions for subject in Mehri transitive constructions, it is not possible to have wh-in-situ in the VSO patterns, as demonstrated in (21a-f).

21a. aĝayg t̲abūr ḍïmah xalfẽt. SVO
the-man broke this window
'The man broke this window.'
mūn t̲abūr ḍïmah xalfẽt? SVO
who broke this window
'Who broke this window?'
ḍïmah xalfẽt mūn t̲abrïs? Topicalized
this window who broke
(This window, who broke?)
t̲abūr aĝayg ḍïmah xalfẽt. VSO
broke the-man this window
'The man broke this window.'
*t̲abūr mūn ḍïmah xalfẽt. VSO
broke who this window
*ḍïmah xalfẽt t̲abrïs mūn? VSO
this window broke who

The transitive V t̲abūr 'broke' selects two arguments in (21a) in SVO; the pre- or post- ver-bal subject, aĝayg 'the man' and the internal object, ḍïmah xalfẽt 'this window'. In the SVO structure, the preverbal subject is replaced by wh-subject mūn 'what', as shown in (21b). This is then topicalized and moved to the [Spec-TopP] projection, as in (21c).

On the other hand, as for the VSO structure in (21d) (which is declarative), the post-verbal subject cannot be replaced by wh-word mūn' who' in (21e-f). This entails that wh-in-situ mechanism is not available in VSO patterns in Mehri; this is evidently shown in the ungram-matical wh-subject constructions in (21e) and (21f). This means that wh-in-situ mechanism is not allowed in VSO patterns in Mehri grammar.

The point that needs to be discussed relates to the pattern for wh-questions in unaccusatives and whether these wh-questions follow wh-movement or wh-in-situ. Let us explain the pattern for wh-questions in unaccusatives and their wh-movement in Mehri in the following examples in (22):

22a..ḥamōh katal baḳa.
the-water spilled out on the ground
'The water spilled out on the ground.'
haśan katal?
What spilled out
'What spilled out?'
hōh katal ḥamōh?
Where spilled out the water
'Where did the water spill out?'
ḥamōh katal hōh?

the-water spilled out where
'Where did the water spill out?'
ḥamōh hōh katal?
the-water where spilled out
'Where did the water spill out?'
katal hōh ḥamōh?
spilled out where the-water
'Where did the water spill out?'

As for the wh-subject in (22a), the definitive subject must be replaced by a wh-word and then is moved overtly to the [Spec-TopP] configuration, while the indefinite subject must remain in-situ in [Spec-TP]. Given the wh-questions, three patterns can be presented: the wh-movement as in (22c) via focalization, the in-situ question as in (22d), which is not focalized, and the partial wh-movement (22e) and (22f) via partial focalization.

Now let us discuss wh-extraction from unaccusative/ergative structures. Given the examples (16–17) above, we show how the wh-prepositional phrase *man-haṡan* 'from what' can appear in wh-interrogatives, as demonstrated in (23) and (24).

23 anīʕāytan ḏa-ḥāyabīt fataṡṡ man-haṡan? SV
the-breasts of-the-camel.3fp.Nom.] [exploded.fp.Perf.] from-what
'What exploded the camel's breasts?'
24 man-haṡan anīʕāytan ḏa-ḥāyabīt fataṡṡ SV
from-what [the-breasts of-the-camel.3fp.Nom.] [exploded fp.Perf.]
'What exploded the camel's breasts?'

(23) and (24) illustrate that the wh-prepositional phrase replaces the original PP *man-aṡxōf* 'from the milk' that causes an explosion to the camel's breasts. The PP *man-aṡxōf* is an adjunct phrase that expands the formation of the core VP [49]. Excluding this PP from the syntactic structure will not affect and harm the meaning of the clause because it does not receive a theta-role from the unaccusative/ergative verb *fataṡṡ* in (16). Moreover, this PP can be replaced by wh-PP *man-haṡan* 'from what', forming two types of interrogatives: a fronting wh-question in (23) and an in-situ wh-question in (24).

Using the standard structure of Chomsky's [10] Phase-based framework in Fig 4 above, let us discuss how the wh-adjunct phrase can be extracted from Mehri's unaccusative/ergative structures, as presented in Fig 5 below.

The syntactic derivation in Fig 5 proceeds in the following manner. The unaccusative/ergative V *fataṡṡ* 'exploded' merges with the DP *anīʕāytan ḏa-ḥāyabīt* 'the camel's breasts' to form the first V-bar. The V-bar in turn adjoins with the adjunct PP *man-aṡxōf* 'from the milk' to form the higher V-bar. The resulting V-bar further merges with the null subject, producing the core VP. This VP is dominated by the defective v*P which lacks an external logical subject. Meanwhile, the lexical V *fataṡṡ* 'exploded' raises to the head T position in order to support its non-lexical features, hence forming the T-bar. Following Chomsky's [1] minimalist analysis, it can be observed in the preceding examples (17) and (23) that the head T contains unvalued agreement features (feminine and singular) and an EPP feature. It probes down searching for the matching goal. Since the external subject is not available at the PF interface, the head T will have a match with the internal DP *anīʕāytan ḏa-ḥāyabīt*, which in turn includes an unvalued case feature. The match is established and the unvalued features on both sides are valued in the course of derivation.

In Chomsky's [10] analysis, head C is viewed as the only category that delivers its probes to the lower functional heads. From Mehri examples (17–23), the head T inherits tense, case, and edge features from C, and overtly attracts DP to [Spec-TP], thus assigning it a nominative case, which is unmarked in Mehri. Assuming the Foc category, it also inherits the edge feature from

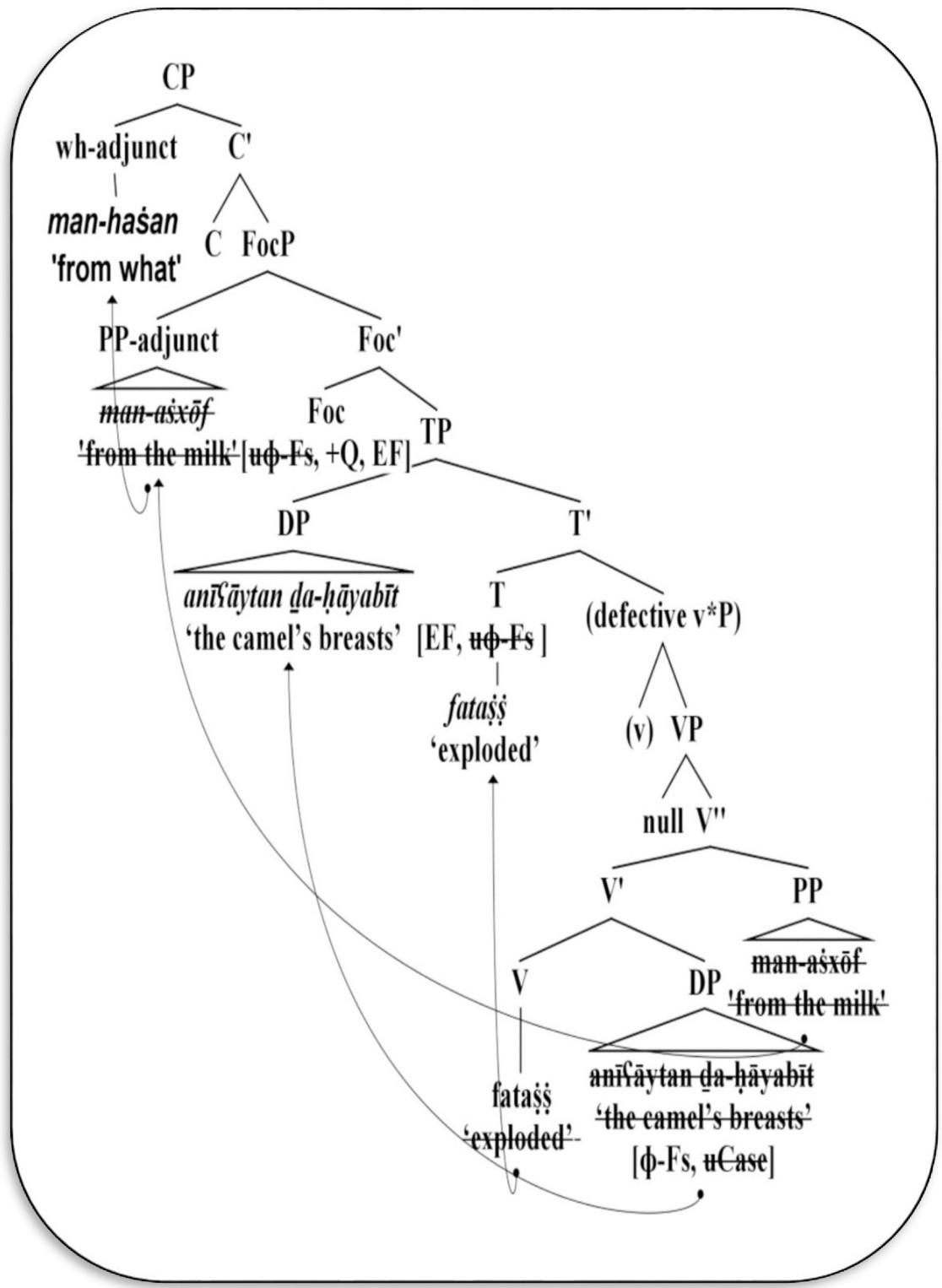

**Fig 5. Derivation of Mehri wh-adjunct questions in unaccusative/ergative structures.**

C. This feature triggers the movement of the adjunct PP *man-aṡxōf* 'from the milk' to [Spec-FocP]. Moreover, Foc contains edge and wh-question features; these features simultaneously seek to replace PP with the wh-prepositional phrase *man-haṡan* 'from what', and then overtly move it to [Spec-CP], as shown in (24). In (24), the head C lacks the edge feature. Therefore, PP *man-aṡxōf* 'from the milk' has to be replaced by the wh-adjunct *man-haṡan* 'from what' and remains in-situ. In both structures, the interrogative force is interpreted within the outer CP phase, where all prior projections have been transferred to PF and LF interface levels.

Let us now discuss wh-subject extraction from Mehri's unaccusative/ergative structures. In (25) and (26) below, the subject wh-question is properly formed, while that in (27) illustrates an *ill*-formed wh-question in the subject position.

25 <u>haṡan</u> fataṡṡ man-aṡxōf?

What [exploded fp.Perf.] [from-the-milk.3ms.Nom.]

'What exploded from the milk?'

26 man-aṡxōf <u>haṡan</u> fataṡṡ?

[from-the-milk.3ms.Nom.] what [exploded fp.Perf.]

'What exploded from the milk?'

27 *man-aṡxōf <u>fataṡṡ</u> haṡan?

[from-the-milk.3ms.nom.] [exploded fp.Perf.] what?

It can be observed in (25–27) that, in the unaccusatives, [PP + V + WHsubj] in (27) is not acceptable, while [PP + V + Subject] is acceptable in (25). The reason that explains this phenomenon is that this type of structure in (27) is not permitted to occur in Mehri grammar, because the adjunct PP *man-aṡxōf* 'from the milk' cannot precede the unaccusatives in the language; that is why the construction in (27) is ruled out in Mehri; adjuncts such as, *man-aṡxōf* 'from the milk' are not allowed to precede the unaccusatives.

Moreover, from (25), the DP *anīʕāytan ḏa-ḥāyabīt* 'the camel's breasts' is replaced by the wh-word *haṡan* 'what'. Since this DP is definite and is located in [Spec-TopP], it can be assumed as a Topic. Hence, the replaced wh-phrase is triggered to move from the external position of TopP to [Spec-CP], as shown in Fig 6.

As demonstrated in Fig 6 the DP *anīʕāytan ḏa-ḥāyabīt* 'the camel's breasts' is originally located within the core VP, namely, it is the complement of the accusative V $\sqrt{f}$ṡṡ 'to explode.' It is triggered to move to [Spec-TopP] because it is definite. It can be observed that the ergative V *fataṡṡ* 'exploded' requires a specifier. Given this, head C inherits a cluster of features to the functional heads Top and T; these features are agreement, case, definiteness, question, and edge features. While head T only inherits agreement features (i.e., unvalued number and gender features) from the head C, it starts probing down, searching for the local matching goal that includes corresponding valued features. This matching is the internal DP *anīʕāytan ḏa-ḥāyabīt* 'the camel's breasts', which bears valued (plural and feminine) agreement features. The head T also lacks an edge feature from C that makes it incapable of extracting DP to the [Spec-TP] position. Instead, the head Top inherits the edge feature, definiteness, and nominative case from C; these features collectively act as the main filter used to value the unvalued case on the DP *anīʕāytan ḏa-ḥāyabīt* 'the camel's breasts', which then moves it to the left periphery of TopP. The extracted DP is considered as the illogical nominative subject for the ergative V *fataṡṡ* 'exploded'. At the same time, the head Top takes both edge and question features from C. The +Q feature thus replaces the extracted DP to a wh-word *haṡan* 'what', while the edge feature moves this wh-word to [Spec-CP] in the outer CP phase. The outcome CP is an interrogative structure that is transferred to PF and LF levels, while the rest projections are fused within it.

Let us explain why the nominative subject is illogical. Case assignment in Mehri is not marked. Unlike Standard Arabic, nominative subject and accusative object in Mehri are

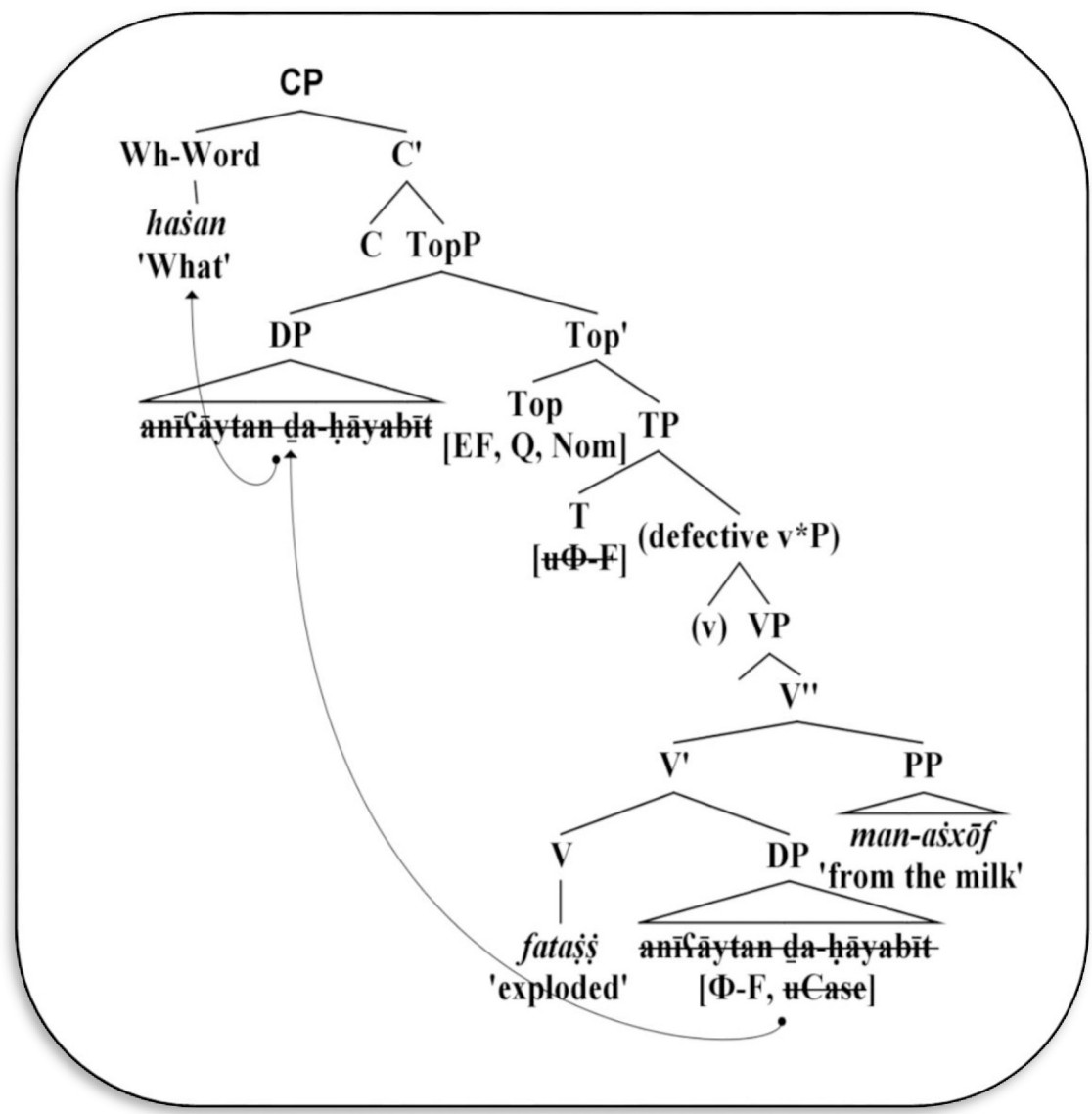

**Fig 6. Derivation of wh-subject extraction in unaccusative/ergative structures.**

syntactically determined. This means they can be recognised through their syntactic positions in a clause. According to Jubilado [60], the unaccusative/ ergative verbs in Philippine languages have only one argument. The one available argument is the one assigned a theme theta role. Similarly, the unmarked nominative subject for the unaccusative/ ergative verbs in Mehri assign only one argument that theta-marks a theme theta role. This argument is not a real subject. It actually works as the internal object for the transitive verb that bears an accusative case. As the verb undergoes some morphological changes, this accusative object is extracted to move to the left periphery of the clause, in which its case is changed to a nominative. Given this, it is considered as an illogical nominative subject, as demonstrated in the following examples (28).

28a. aĝayg ṣikk axalfẽt

the-man closed the-window

'The man closed the window.'
b. axalfē̆t ṣatakūt
the-window closed
'The window closed.'

Similarly, the extracted DP *anīʕāytan ḏa-ḥāyabīt* 'the camel's breasts' in (26) is replaced by the wh-word *haṡan* 'what', which remains in [Spec-TopP], while the adjunct PP *man-aṡxōf* 'from the milk' undergoes movement to the left periphery of FocP below CP. In this case, the edge feature on the head Foc, inherited from C, motivates the adjunct PP *man-aṡxōf* to move to [Spec-FocP], as illustrated in Fig 6 above. On the contrary, the head Top lacks to inherit an edge feature from C. Therefore, the replaced wh-word *haṡan* appears in [Spec-TopP]. The structure still shows an interrogative force because C bears the +Q feature that is inherited by Top, replaces the extracted DP, and covertly moves its formal question features to [Spec-CP] for the reason that the edge feature on the head Top is unavailable.

Regarding the *ill*-formed structure in (27), it can be argued that the ergative verbs in Mehri cannot show VS word order because the extracted DP is an illogical subject, which is originally located in the internal verbal position. However, moving V *fataṣṣ* 'exploded' over DP *anīʕāytan ḏa-ḥāyabīt* 'the camel's breasts'–replaced by wh-word *haṡan* 'what'–is syntactically banned in Mehri syntax. It is obvious that the core VP is incomplete to fuse within the v*P phase. Thus, there is a defective v*P that does not include a real external argument. The VP also is incomplete because it does not project a light v that acts as the accusative case assigner.

## Wh-phrase extraction from unergative structures

Traditionally, verbs that only contain one argument are termed intransitive verbs [6, 56]. Based on the views of generative linguists, the intransitive verb type also has a subtype called an unergative verb, which is exactly the opposite of the ergative type. It has an external true subject that theta-marks an agent or experiencer theta-role. This type creates a v*P phase because the light v is abstractly available, which covertly inherits the accusative case to the null verbal complement. For example, the verbal root $\sqrt{syr}$ 'to go/walk' is an unergative type. This verb obligatorily selects an experiencer subject NP, whereas its internal locative goal is null. Semantically, the Mehri verb $\sqrt{syr}$ 'to go/walk' denotes motion that is performed by a real experiencer subject, and a typical destination is syntactically invisible at the PF interface [61]. For further exemplification of the unergative structures, consider (29) and (30).

29 ḥāybīt ḥōfay tašaḥrawd (man masyer)
[the-camel fs.milked fs.Nom.] [become-tired.3fs.Subj] (from-the walk)
30 tašaḥrawd ḥāybīt ḥōfay (man masyer)
[become-tired.3fs.Subj] [the-camel.fs.milked.fs.Nom.] (from-the walk)
'The milked camel will become tired (from the walk).'

Lexically, the predicate *ša-ḥrawd* 'become tired' is a *ša*-type verb [56]. This verb is conversely derived from the simple triliteral verb $\sqrt{ḥrd}$ 'to be strong'. In (29) and (30), the verb *tašaḥrawd* 'become tired' agrees with the preverbal or post-verbal DP ḥāybīt ḥōfay 'milked camel' in third person, feminine and singular features. This verb is an unergative type because it assigns only one argument, whereas the bracketed PP *(man masyer)* 'from the walk' is merely an adjunct. It can be observed that (29) is an SVO word order in which the preverbal DP ḥāybīt ḥōfay 'milked camel' occupies [Spec-TopP]. On the other hand, (30) is a VSO order wherein the post-verbal DP ḥāybīt ḥōfay 'milked camel' remains in-situ within the v*P phase and occupies [Spec-v*P]. Concerning argument/thematic structure, Jubilado [60] indicated that unergative verbs only assign one argument, which theta-marked an agent or experiencer theta-role. Similar to this, the unergative verb *tašaḥrawd* 'become tired' in both structures

assigns only one argument in (29) and (30); it is the pre-verbal or post-verbal DP ḥāybīt ḥōfay 'milked camel'. This DP semantically shows an experiencer theta-role associated with the theta-grid tašaḥrawd 'become tired'. In (29) and (30), the selected DP is an active goal that contains an unvalued case. Therefore, the functional heads, Top and T, inherit the case from C and covertly assign the nominative case to the selected DP ḥāybīt ḥōfay 'milked camel.'

Let us now look at the patterns of the wh-questions in Mehri unergatives, as demonstrated in (31) below. There are no differences with regard to wh-questions in the unaccusatives and those in the unergatives in Mehri because both are intransitive types, which need to assign only one argument NP. Both of them pattern with wh-questions in active constructions, as shown in the sections on unaccusatives and unergatives.

31a. Ĝagnūt šawkfūt brak aṣayga
the-girl slept in the-cave
'The girl slept in the cave.'
b. mūn šawkfūt brak aṣayga?
who slept in the-cave
'Who slept in the cave?'
c. hōh šawkfūt Ĝagnūt?
where slept the-girl
'Where did the girl sleep?'
d. Ĝagnūt šawkfūt hōh?
the-girl slept where
'Where did the girl sleep?'
e. Ĝagnūt hōh šawkfūt?
the-girl where slept
'Where did the girl sleep?'
f. šawkfūt hōh Ĝagnūt?
slept where the-girl
'Where did the girl sleep?'

The (V)erb √škf "to sleep" is intransitive, specifically, it is an unergative type that assigns one argument; it is an experiencer subject. In (31a), the V šawkfūt 'slept' assigns an experiencer subject NP, Ĝagnūt 'the girl', while the PP brak aṣayga 'in the cave' is an optional locative goal that does not affect the meaning, if deleted. Based on the (in)definiteness, the wh-subject can either move to [Spec-TopP], as in (31b) or to [Spec-TP]. As for the wh-object question, the wh-word undergoes movement to the left periphery of the clause via the head Foc and EF inherited from ForceP. However, three wh-patterns can be observed: the front (overt) wh-movement, as in (31c), the in-situ pattern, which is not focalized, as in (31d), and the partial wh-movement, as in (31e) and (31f).

With regard to extracting wh-phrases from unergative structures, it is noticed that the wh-subject phrase in (32) is grammatically formed, while (33) is ill-formed; the interpretation in the latter is being violated because it can be claimed that the subject wh-phrase must move overtly in SV order, while it should be banned in VS order.

32 haṣan tašaḥrawd (man masyer)?
what [become.3fs.subj] (from the walk)
'What will become tired from the walk?'
33 *tašaḥrawd haṣan (man masyer)?
[become-3fs.subj] what (from the walk)

In (33), the unergative verb has been intonated more than the wh-subject. This makes the interrogative clause lose its question force. Therefore, the unergatives [V + WHsubj + PP] is

not acceptable in Mehri. Furthermore, wh-subject movement is not allowed in VS word order in Mehri, and this explains why (33) is ungrammatical. On the other hand, the pattern [V + WHsubj + Object] is not acceptable in transitive constructions in Mehri, as demonstrated in the following ill-formed example in (34a).

34a *ṯabūr mūn aśafrḗt?
broke who the knife
'Who broke the knife?'
b. mūn ṯabūr aśafrḗt?
who broke the knife
'Who broke the knife?'

The pattern [V + WHsubj + Object] is not acceptable in transitive constructions in Mehri in (34a). In (34b), the wh-word *mūn* 'who' replaces the post-verbal subject in the VSO structure and then moves higher up in the clause structure. To construct a correct wh-subject question, the post-verbal subject must subsequently move to the [Spec-TP] projection, until reaching the final locus in [Spec-TopP]. Moreover, the extracted subject can be substituted by the wh-word *mūn* 'who' and moves to [Spec-ForceP], which results in an interrogative force.

Moreover, in (29–30), the experiencer DP ḥāybīt ḥōfay 'milked camel' underwent overt movement from [Spec-v*P] to [Spec-TopP]. Since the head Top inherits the Q-feature and edge feature from the head C, the relevant DP will be replaced by the wh-subject *haṣan* 'what' and then will move to the left periphery of CP, thus forming a grammatical interrogative. Consider Fig 7 below.

As shown in Fig 7 above, the wh-subject question in (29–30) shows the structural projection of the unergative verb *tašaḥrawd* 'will become tired'. This verb is drawn from the lexicon with a bundle of features such as a Φ-feature (3mf.), subjunctive and causative features. However, this verb initially raises to adjoin with the affixal light v in order to support its non-lexical features, forming a verbal complex that includes a causative feature. After spell-out, the outcome complex incorporates a PP-adjunct causer *man masyer* 'from the walk' under Merge operation in the sense that Merge operation gives the lexical core VP *tašaḥrawd man masyer*, which, in turn, merges with the light v to form the light v-bar and then receives a spell-out in the phonological level. Consequently, the resulting v-bar merges with the DP specifier ḥāybīt ḥōfay 'the milked camel' to form an inner v*P phase. What makes it a phase in Fig 7 is the fact that the light v has an external, real experiencer subject such as ḥāybīt ḥōfay 'the milked camel', while the ergative verb in Fig 6 possesses an illogical, extracted subject such as *aniˁāytan ḏa-ḥāyabīt* 'the camel's breasts'. As a result, the light v (including lexical V) in Fig 7 is the head of the phase v*P, which denotes a complete thematic/argument meaning to the given verb, whereas the core VP lacks to form a v*P phase in Fig 6.

Regarding the TP layer, the functional head T is further selected; it contains non-lexical features such as an unvalued Φ-feature (3mf.) and temporal features (Subj.). Therefore, it is an active probe that starts to probe down searching for the local goals. The lexical V must undergo movement to the head T position in order to justify its temporal and aspectual features [36, 61–63]. The probe T is strong and triggers movement of the complex v *tašaḥrawd* 'will become tired', thus forming the subjunctive verb. Moreover, T inherits the unvalued Φ-Feature (3mf.) from the head C, which enables it to agree with the closest DP ḥāybīt ḥōfay in v*P phase that bears corresponding valued features, namely third singular and feminine. This DP (ḥāybīt ḥōfay) remains in situ because T lacks an edge feature from C, thereby [Spec-TP] configuration remains null at PF.

In relation to the TopP layer, the functional head Top is selected. It inherits a bundle of features such as edge, definite, nominative, and question features from source C. Since [Spec-v*P] ḥāybīt ḥōfay contains an unvalued case feature that requires valuing, the head Top acts as

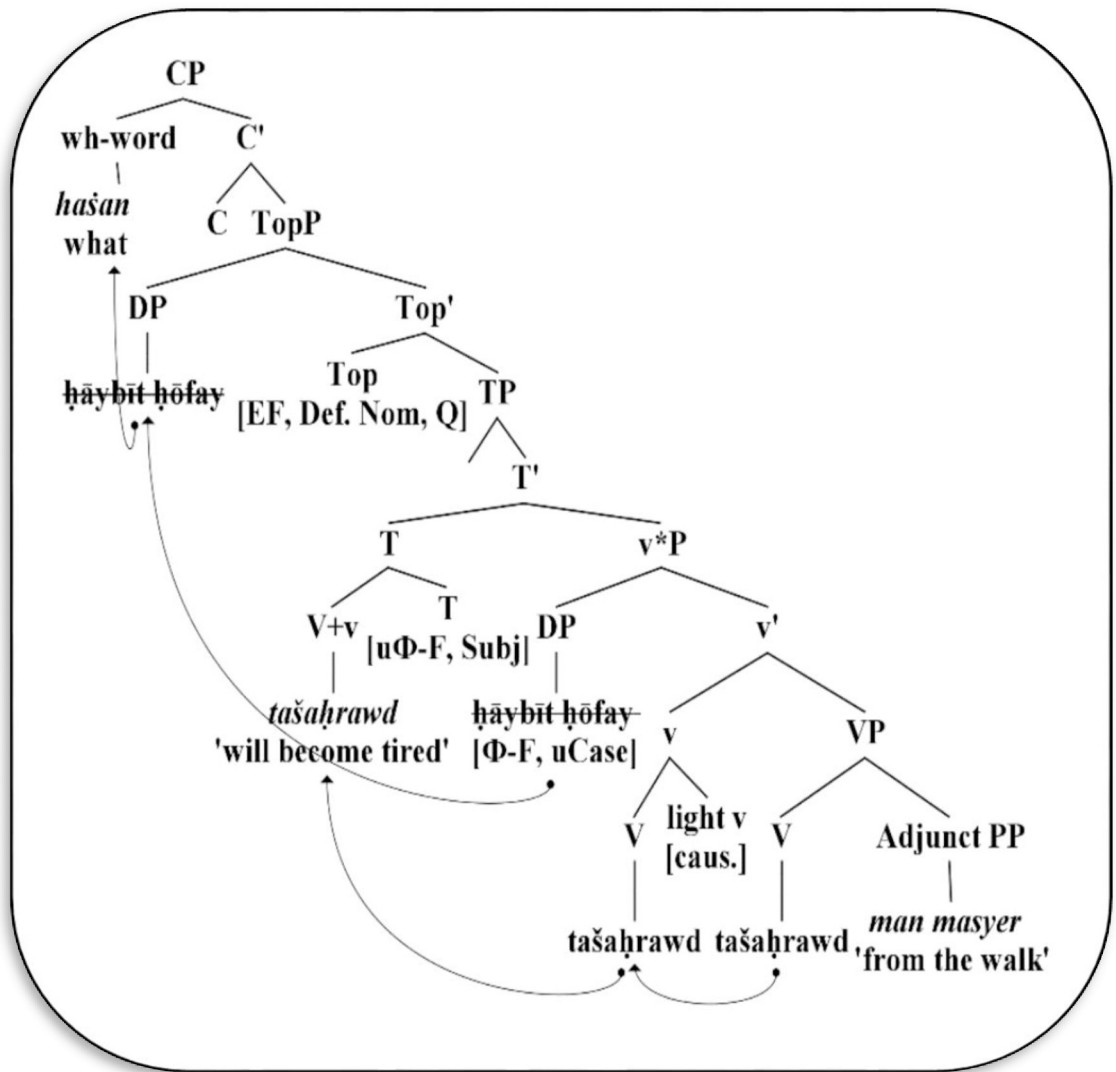

**Fig 7. Derivation of wh-subject question in unergative structures.**

the case assigner because it inherits definite and nominative case features from C. Given this, Top also takes an edge feature from C in order to attract [Spec-v*P] ḥāybīt ḥōfay to the left periphery of TopP projection, thus rendering it a topic rather than a subject. Not only this, but the head Top also inherits a +Q feature from C and hence replaces the Topic ḥāybīt ḥōfay 'the milked camel' with the wh-subject haṡan 'what', which overtly moves to [Spec-CP] via the edge feature. The outcome CP is the outer phase that is transferred to PF and LF interface in order to give an interrogative force to the structure, while the complemental projections TopP, TP and v*P phase receive null spell-out at PF.

In the case of the *ill*-formed structure in (33), the wh-formation is not allowed in the VS order in Mehri syntax, because the temporal complex *tašaḥrawd* 'will become tired' is banned from crossing over the wh-word *haṡan* 'what' in unergative structures. The given structure in the unergative type gives further support to the argument that the wh-subject phrase in Mehri is an obligatory operation that takes place in overt syntax, otherwise, the derivation will crash, as illustrated in (33), where the wh-word *haṡan* 'what' remains in situ within v*P phase.

On the other hand, the wh-adjunct questions in unergative structures are shown in (35) and (36); the wh-adjunct *man-haṡan* 'from what' appears in two different positions: in the right periphery and left periphery of the clauses. This logical contrast allows us to argue that the wh-adjunct movement in Mehri's unergative structures is optional. This entails that the wh-adjunct phrase can be fronted to the clause-initial position (that is, the outer CP phase), as in (35), or remains in situ, as in (36).

35 ḥāybīt ḥōfay tašaḥrawd man-haṡan
[the-camel-fs-milked fs.Nom.] [become-tired.3fs.Subj] (from-what)
'Lit. From what will the milked camel become tired?'
'What will make the camel tired?'
36 man-haṡan tašaḥrawd ḥāybīt ḥōfay
(from-what) [become-tired-3fs.Subj] [the-camel.fs-milked.fs.Nom]
'Lit. From what will the milked camel become tired?'
'What will make the camel tired?'

For further explanation, Fig 8 below supports the examples (35) and (36) above. The tree diagram illustrates the motives of extracting the wh-adjunct phrase in the clause-initial position or remaining in-situ in unergative structures.

In Fig 8, two propositional phases are available: the v*P phase represents the argument/thematic structure of the unergative verb *tašaḥrawd* 'will become tired' and the outer CP phase shows the interrogative force of the outcome structure. Regarding the v*P phase, the lexical V *tašaḥrawd* raises to adjoin the light v that bears the causative feature. The outcome complex (V +v) further merges with the PP complement *man masyer* 'from the walk'. The complex verb crucially requires an external argument. It thus merges with the DP *ḥāybīt ḥōfay* 'the milked camel', which itself theta-marks the experiencer theta role. As a result, the v*P phase is formed, while the lexical VP receives a null spell-out at PF and fuses within the v*P phase.

Moving to the CP phase, similar to the analysis in Fig 7 the verbal complex *tašaḥrawd* raises to adjoin the head T because the latter head is strong and needs to verify its temporal (subjunctive) features. Moreover, the head T inherits unvalued Φ-features (3mf.), which enable it to agree with [Spec-v*P] *ḥāybīt ḥōfay* 'the milked camel' that bears intrinsic corresponding features, and hence, the unvalued features on T will be valued and then deleted in the syntax. The head T also lacks an edge feature, which means the matching DP *ḥāybīt ḥōfay* remains within v*P. Instead, the head Top is selected and consequently inherits edge, definite, and nominative features from C. While the definite and nominative features act as the nominative case assigner on DP, the inherited edge feature on Top triggers movement of the DP *ḥāybīt ḥōfay* to the left periphery of TopP.

In the context of wh-adjunct questions in unergative structures, it is argued that the head Foc optionally inherits the edge feature from C, in order to attract the adjunct to [Spec-CP]. In other words, the head Foc inherits a +Q feature from C which may include the edge feature that has the virtue of replacing the PP-adjunct *man masyer* 'from the walk' to the wh-adjunct phrase *man-haṡan* 'from what', and then it moves overtly it to [Spec-FocP] and then to [Spec-CP], as demonstrated in Fig 8 and example (36). On the contrary, the head Foc may lack the edge feature from C. Therefore, it covertly moves formal features of the adjunct to [Spec-CP], whereas the lexical wh-adjunct remains in situ within v*P, as illustrated in (35). This argument leads us to propose that the wh-adjunct phrase movement in Mehri unergative structures is an optional operation in the sense that the wh-phrase can be fronted to the clause-initial position or remains in situ in the syntax. That is, Mehri allows two types of wh-adjunct question strategies: wh-adjunct fronting and wh-in situ.

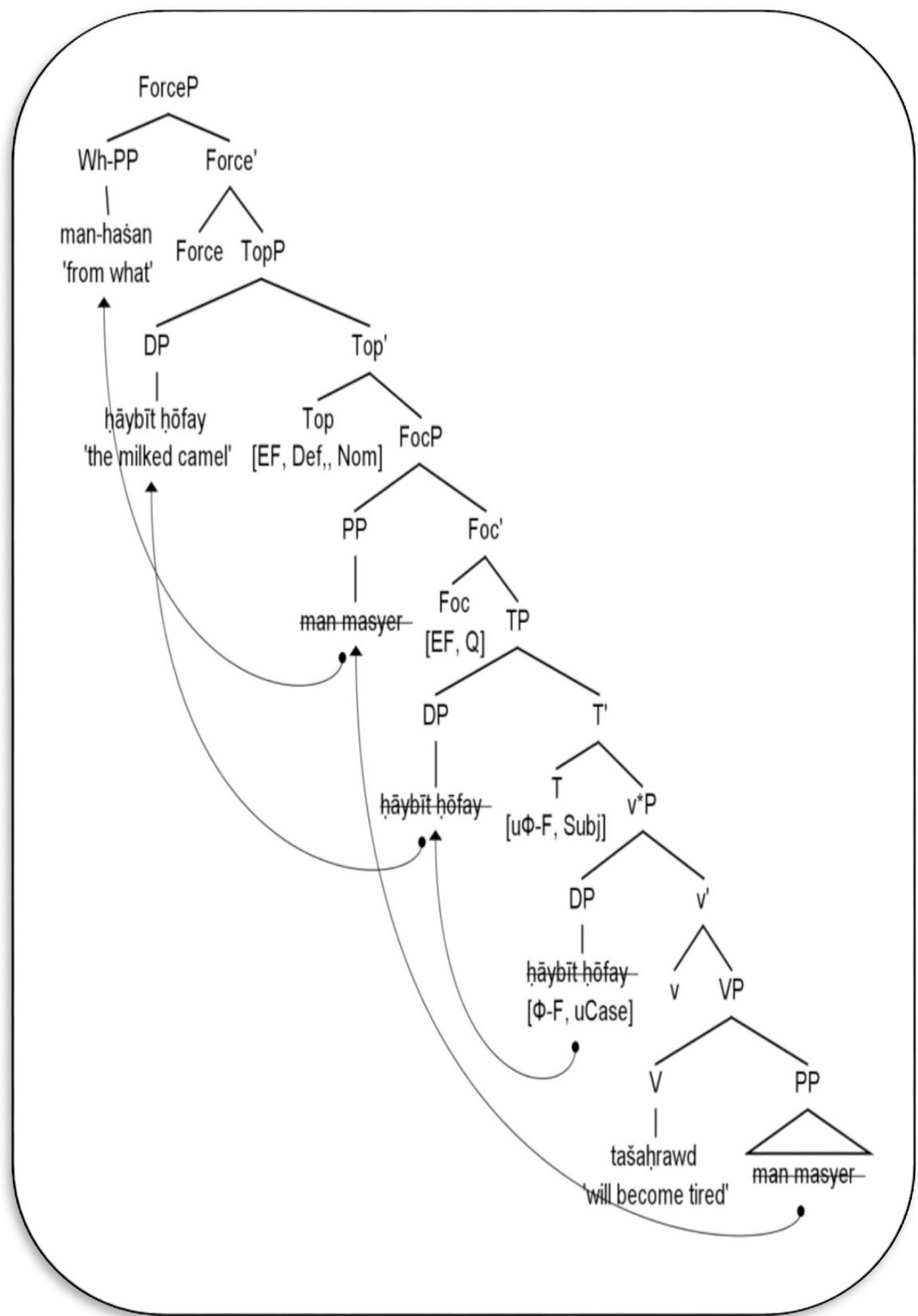

**Fig 8. Derivation of wh-adjunct questions in Mehri unergative structures.**

## Conclusion

This study revealed that the Mehri unaccusative/ergative verb is morphologically marked by the infix [-*ta*-]. This verb lacks a causative/transitive feature that makes it unable to split into a light v, and contains an illogical thematic subject, henceforth v*P is defective and cannot be considered as a single phase. On the other hand, the Mehri unergative verb obligatorily has a logical external specifier that theta-marks an agent or experiencer theta role subject; this subject merges with the light v, which is the head of v*P. Furthermore, the paper examined the interaction between the recent minimalist analysis of [10, 11] Phase-based approach and Mehri wh-questions, it provided further support to Chomsky's minimalist approach. It investigated the functional head C in Mehri and showed that it is the main source that provides edge and question features to the head Foc for the wh-adjunct questions and the head Top for the wh-subject question.

Moreover, the current study presented an adequate answer to the research questions; it explained that Mehri language permits fronting of wh-phrases to [Spec-CP]? and provided a satisfactory account of wh-question movement in Mehri unaccusative and (un)ergative structures within Chomsky Phase-based Theory.

In the extraction of wh-adjunct phrases, both ergative and unergative verbs can overtly and covertly exhibit wh-movement. This means that once the head Foc inherits the edge feature from C, the wh-adjunct must overtly move from its original position within v*P to [Spec-FocP] and subsequently to [Spec-CP]. On the other hand, the lexical wh-adjunct has to remain within v*P whereas its question features covertly move to [Spec-CP] because the head Foc does not inherit an edge feature from C.

Concerning the wh-subject question, the wh-subject overly employs movement to [Spec-CP] because the abstract C obligatorily inherits the edge feature to the head Top, which triggers movement of the illogical subject in ergative verb structures and the logical external specifier in unergative structures to the left periphery of CP phase. Besides, in both Mehri ergative and unergative structures, the wh-subject movement can be allowed only with SVO. As a result, two assumptions can be made here: (i) In ergative structures, the wh-subject does not represent a real argument. That is, the movement of an ergative verb over the extracted subject to form a VS order is syntactically banned. (ii) In unergative structures, the wh-subject represents a real argument that originally occupies [Spec-TP] or [Spec-Top]; this argument has to move overtly to [Spec-CP], otherwise, the derived structure will crash in the course of derivation if the unergative verb moves to T and the replaced wh-subject remains in-situ within v*P.

Furthermore, this study provides theoretical implications; the findings of the study provide further support to Chomsky's Phase Theory, where the analysis has shown that Mehri obeys the Phase Impenetrability Condition proposed in [10], which assumes that when all syntactic operations in a given phase have been achieved, the domain of that phase becomes impenetrable to any syntactic operation in the syntax. Another implication is that, unlike the assumption of [64], which states that ergativity in Aramaic dialects is unique in Semitic tradition due to the contact with ergative languages such as Kurdish and Iranian languages, this article demonstrated that the ergative morphology is invariably attested in Mehri, which belongs to the Semitic language family. It is also hoped that the syntactic findings of the current study will contribute to the understanding of the syntax of wh-questions in the Semitic language family and the existing literature on the Arabic language and its varieties as well as on other languages in the world.

## Author Contributions

**Conceptualization:** Abdul-Hafeed Fakih.

**Data curation:** Saeed Saad Al-Qumairi.

**Formal analysis:** Saeed Saad Al-Qumairi.

**Investigation:** Saeed Saad Al-Qumairi.

**Methodology:** Saeed Saad Al-Qumairi.

**Project administration:** Saeed Saad Al-Qumairi.

**Resources:** Saeed Saad Al-Qumairi.

**Supervision:** Abdul-Hafeed Fakih.

**Validation:** Abdul-Hafeed Fakih.

**Visualization:** Abdul-Hafeed Fakih.

**Writing – original draft:** Saeed Saad Al-Qumairi.

**Writing – review & editing:** Ali Abbas Falah Alzubi.

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
