## [Decision Letter · Decision Letter 0]

11 Apr 2023

PONE-D-23-09811The Syntax of Wh-Questions in Unaccusative and (Un)Ergative Structures in Mehri Language: A Phase-Based ApproachPLOS ONE

Dear Dr. Alzubi,

Thank you for submitting your manuscript to PLOS ONE. After careful consideration, we feel that it has merit but does not fully meet PLOS ONE’s publication criteria as it currently stands. Therefore, we invite you to submit a revised version of the manuscript that addresses the points raised during the review process.

We look forward to receiving your revised manuscript.

Kind regards,

Thiago P. Fernandes, PhD

Academic Editor

PLOS ONE

Journal Requirements:

   "The authors are thankful to the Deanship of Scientific Research at Najran University for funding this work, under the General Research Funding Program Grant Code (NU/DRP/ SEHRC/12/3)"

   "Yes, the funders had no role in the study design, data collection and analysis, the decision to publish, or the preparation of the manuscript."

Additional Editor Comments:

Please enlist a native English speaker to edit your manuscript and also check references - some nonmentioned and some incorrect accordingly to the standards.

Thank you for submitting your valuable work.

The reviews, which are insightful and interesting, pointed to some unexplained aspects. The authors will notice the reviewers found merits in your protocol, but also raised several important concerns.

If you are willing to undertake the recommended points, I would be pleased to reconsider the manuscript. If you disagree with any of the recommendations, I would also be willing to consider a rebuttal to any of the points made. For your guidance, the comments obtained during review are appended below. We hope that you find these helpful.

It is important to note that we cannot make any promises as to whether your updated manuscript will be considered and you should consider the extent to which you can address the comments below before resubmission.

If you choose to revise the work, please submit a detailed list of changes for each point raised when you submit the revised manuscript. Please also highlight where the text has been changed in the resubmitted file - this will help to streamline the reviewing process and minimise any delays.

Reviewers' comments:

Reviewer's Responses to Questions

**Comments to the Author**

1. Is the manuscript technically sound, and do the data support the conclusions?

Reviewer #1: Yes

Reviewer #2: Partly

2. Has the statistical analysis been performed appropriately and rigorously? 

Reviewer #1: N/A

Reviewer #2: N/A

3. Have the authors made all data underlying the findings in their manuscript fully available?

Reviewer #1: Yes

Reviewer #2: Yes

4. Is the manuscript presented in an intelligible fashion and written in standard English?

Reviewer #1: Yes

Reviewer #2: No

5. Review Comments to the Author

Reviewer #1: The analysis is fine, and the authors have provided a description of wh-movement in unaccusatives and unergatives in the Southern Arabic language Mehri. As the authors simply aim to document this language under the generative framework and do not aim to provide new insights into generative linguistic theories, I find the stated objectives to have been reached.

However, the manuscript needs improvements in a few places. The authors are advised to bear in mind that not all readers are familiar with the Arabic language(s) and the terms used in the manuscript. I will point out a few of these which can be confusing sometimes:

1. The authors's description of Mehri seems to imply that this language is not a type of Arabic (e.g., "is threatened by the majority language, Arabic", until they later mentioned that it is a Southern Arabic language. This makes me very confused when the authors have a whole section on "wh-constructions in Arabic" which initially strikes me as irrelevant to Mehri. Thus, it is very important for the authors to be clear in the beginning that Mehri IS an an Arabic language. Also, which variety of Arabic languages do the authors have in mind when comparing "Arabic" to Mehri?

2. When mentioning the terms "unaccusative" and "unergative", please provide examples (maybe place them in brackets like e.g., 'xxxxxx'). The authors only mentioned the examples on p.11, which is too late.

3. On p.6, reference [22] mentioned that preverbal DPs are base-generated in spec-TopP. The authors' description seems to be that subjects are not base-generated. They subsequently move to the spec-TopP. The authors did not explain this contradiction. Do the authors agree with or disagree with reference [22]?

4. There are a lot of typos in this manuscript. For example, the glossing in (4) on p.6 is incorrect. The word order is OVS ('what ate Ahmed?'). I also see frequently "Error! Reference source not found" on p. 14-15. This causes troubles for me when I try to understand what the authors try to convey. There are other typos which I will not list here. Please correct them.

5. Please explain what "radical roots" means on p.13 (including the terms like "trilateral"). Some readers – especially people not working on Arabic – are not familiar with such terms.

6. The figures look very crowded. Please make the pictures bigger and tidier. When I read the detailed descriptions, I sometimes found it easier to simply follow the text than check the figure.

7. Although the authors do not aim to provide unexpected data to make a new point for the field, it would be nice if the authors could mention or add some implications of their analysis for the field, other than simply confirming the established phase theory. If a linguist not working on this language reads this paper, what do they gain?

Reviewer #2: In this manuscript, the authors provide an analysis of wh-movement from in unaccusative and unergative constructions in Mehri. I do not recommend accepting the manuscript in its current form.

First of all, the manuscript should be carefully proofread before future re-submission. Currently, it is filled with missing/wrong references to examples and figures. For example, there are more than one figures labeled figure 1, same with figure 2; on page 7 just below figure 1, there is an in-text figure reference without any numbering; on pages 14-16, there is something wrong with example referencing (it reads “Error! Reference source no found” in text several times); on page 15, there are multiple references to a non-existing example “0”. In addition to all these errors, there are also a lot of typos, e.g. the word “noble” should be “novel” in the abstract, the word order in the gloss of example (4) is wrong, there is a sentence towards the middle of page 18: “… remains in-situ in.” which is either unfinished, or contains an extra “in”, etc.. All these problems are extremely distracting, and make the manuscript hard to read.

Here are some more substantive issues:

- Section 1: This is a rather lengthy discussion on wh-movement in Arabic. This whole discussion has very little to do with how wh-movement works in Mehri, which I assume is the main point of this paper. Just because Arabic and Mehri are related languages does not mean that the authors can make syntactic assumptions about Mehri based on what has been established in Arabic. I recommend either removing this discussion entirely, or make clear what this discussion has to do with the rest of the paper.

- p.12: I recommend having a tree demonstrating the split-CP hypothesis, which is implicitly adopted throughout the manuscript (the ForceP-FocP-TopP left periphery). This is worth making clear, since the ordering of FocP and TopP is doing a lot of the heavy-lifting in accounting for examples like (11) and (12) later on.

- Middle of p.13: “Chomsky stressed that when all syntactic operations in a given phase have been completed…” Merger of the phase head triggers the spell out of phasal complement. Merger of phase head is crucially different from “when all syntactic operations in a phase are completed”.

- Bottom of p.17: “Meanwhile, the lexical V ‘exploded’ raises to the head T position in order to …” You haven’t yet established whether/how V-to-T head movement in Mehri works (observing that it happens in Arabic is not a justification that the same happens in Mehri). Please provide justification that V-to-T movement happens in this case, or state clearly that this is an assumption you are making (rather than stating it as part of your proposal).

- Bottom of p. 17: “… it can be observed in the preceding examples (8) and (9) that the head T contains unvalued agreement features and an EPP feature.” How do (8) and (9) show you that T has an EPP feature? (8) and (9) only shows that the subject DP precedes the V. How do you know that the subject DPs are in Spec-TP (as opposed to, for example, Spec-TopicP)?

- More generally, how do we know that (9) is an instance of overt wh-movement, rather than topicalization of an in-situ wh-PP? Or maybe just base-generation of the wh-PP at a high position? One idea is to test whether this movement is island sensitive (and if so, that rules out the base-generation possibility). You will still need some independent evidence to rule out the topicalization possibility.

- p. 18: “From Mehri examples (8-9), the head T inherits tense, case, and edge features from C…” Please be clear how the two sentences in (8) and (9) show this (especially, that T inherits EF from C).

- Top of p.19 “Since this DP is definite and is located in Spec-TopP, it can be assumed as a Topic”. Why is it in Spec-TopP, as opposed to Spec-TP? No independent evidence is provided. Some diagnostics for structural height could be helpful here (e.g. ordering between the DP and certain TP-adjuncts).

- bottom of p.19: “… it is triggered to move to Spec-TopP because it is definite.” Why would definiteness trigger movement to Spec-TopP in Mehri? Right now the wording makes it sounds like all definite DPs in Mehri need to undergo Topic movement (which can’t be right). Is there independent evidence for this?

- Is there independent evidence that wh-DPs can be licensed in-situ in Mehri?

- p.21: traditionally, experiencer arguments are thought to be VP-internal arguments (Landau 2010). Under this view, wouldn’t “become.tired” actually be projecting “camel” VP-internally unlike an unergative verb? Why not just use clearly unergative verbs like “walk” or “swim” in your analysis instead?

- middle of p.25 “… because the temporal complex is banned from crossing over the wh-word…” Why is V-to-T movement blocked by the wh-word? There is no explanation provided.

- Is the unergative counterpart of example (11) possible? Under your analysis, if the adjunct “man masyer” is moved to Spec-Foc, the wh-DP could be licensed in-situ (maybe in Spec-TopP) with only covert movement of a Q-feature. This predicts a sentence like “man masyer haṡan tašaḥrawd” to be possible.

- What about multiple-wh questions (e.g. Who ran to which house)? Is that a possibility in Mehri? Either way, it would be a nice testing bed for the proposed theory.

6. PLOS authors have the option to publish the peer review history of their article (what does this mean?). If published, this will include your full peer review and any attached files.

Reviewer #1: **Yes: **Jun Lyu

Reviewer #2: No

---

## [Author Response · Author response to Decision Letter 0]

1 May 2023

Response to Reviewer #1

1. The authors' description of Mehri seems to imply that this language is not a type of Arabic (e.g., "is threatened by the majority language, Arabic", until they later mentioned that it is a Southern Arabic language. This makes me very confused when the authors have a whole section on "wh-constructions in Arabic" which initially strikes me as irrelevant to Mehri. Thus, it is very important for the authors to be clear in the beginning that Mehri IS an Arabic language. Also, which variety of Arabic languages do the authors have in mind when comparing "Arabic" to Mehri?

The answer to comment 1 is provided in the text of the paper (See the introdcution).

As a sub-group of the Semitic language family (which is a branch of Afroasiatic family), Mehri is the most spoken of the Modern South Arabian languages (MSALs) in the present Arabian Peninsula. In other words, Mehri is a Southern Arabic language. Before the advent of Islam, Mehri and its sister MSALs were spoken largely in the southern part of the Arabian Peninsula. Moreover, traditional and modern Arabic scholars, who worked on Arabic dialects, stated that Mehri is not only a branch of the Southern Eastern Arabic language spoken in Al-Mahra region and Socotra in Yemen, but also an old Arabic tongue.

2. When mentioning the terms "unaccusative" and "unergative", please provide examples (maybe place them in brackets like e.g., 'xxxxxx'). The authors only mentioned the examples on p.11, which is too late.

The answer to comment 1 is provided in the text of the paper as an endnote. This endnote is as follows: 

“Unaccusative/ergative and Unergative verbs are one-place predicate. They are traditionally called intransitive verbs. Unaccusative/ergative verbs in Mehri language are derivatives in which the external argument is not a true subject. They are derived from accusative verbs. The affixed element -ta- (attached to verbs) is the unaccusative marker. This can be exemplified in Table (1). 

Accusative

Two-place predicate Unaccusative

One-place predicate

 Gloss

fiṩṩ 

ḥiṣṣ

riṩṩ

ķiṩṩ

raḥāṩ

ḥamōṩ fataṩ 

ḥataṣ

rataṩ

ķataṩ

rataḥaṩ

ḥatamaṩ explode

roughly tie

partly break

damage 

wash

sour

Contrastively, the unergative verbs select an external true subject that theta-marks an agent or experiencer theta-role. This can be demonstrated in Table (2): 

Unergative 

One-place predicate Gloss

šaḥrōd 

sayūr

ḏalūf

baķūṩ 

šawkōf

śaxawlōl become tired 

go

jump

run

sleep

set down 

3. On p.6, reference [22] mentioned that preverbal DPs are base-generated in spec-TopP. The authors' description seems to be that subjects are not base-generated. They subsequently move to the spec-TopP. The authors did not explain this contradiction. Do the authors agree with or disagree with reference [22]?

The authors think that there is no contradiction in this assumption. Both Mehri and Arabic do have SVO and VSO. The preverbal DPs are triggered to move to the spec-TopP configuration via edge and topic features, which are inherited from C.

4. There are a lot of typos in this manuscript. For example, the glossing in (4) on p.6 is incorrect. The word order is OVS ('what ate Ahmed?'). I also see frequently "Error! Reference source not found" on p. 14-15. This causes troubles for me when I try to understand what the authors try to convey. There are other typos which I will not list here. Please correct them. 

The answer to comment 4 is provided in the text of the paper. The glossing (4) on p. 6 has been corrected. “(The word order is OVS ('what ate Ahmed?')” has been corrected. "Error! Reference source not found" on pp. 14-15” has been corrected. 

5. Please explain what "radical roots" means on p.13 (including the terms like "trilateral"). Some readers – especially people not working on Arabic – are not familiar with such terms.

The authors have added an endnote (in the end of the paper) that explains what “radical roots” mean. It also illustrates what the terms “trilateral roots” and quadrilateral roots” mean. This endnote is as follows:

“In Semitic languages, the verb roots are indicated by a sequence of consonants (dubbed as radicals); this invites the terminology ‘consonantal root.’ In Arabic, for instance, consonantal roots are used to form words by the addition of vowels and non-root consonants. There also are other terms used in Arabic such as, “trilateral roots” and quadrilateral roots”. A trilateral root (dubbed as ʻtriconsonantal rootʻ) is viewed by Arab and Western scholars as a root that has a sequence of three consonants (e.g., k-t-b ك-ت-ب "to write"). A quadrilateral root, on the other hand, is a consonantal root that has a sequence of four consonants. A quadriliteral form is a word derived from four consonantal roots in Arabic (e.g, the quadriliteral root عرقل ʻto obstructʻ).”

6. The figures look very crowded. Please make the pictures bigger and tidier. When I read the detailed descriptions, I sometimes found it easier to simply follow the text than check the figure.

The authors have made the tree-diagrams bigger and tidier in the text of the paper; we have changed those which are not clear and have also enlarged the small ones. 

7. Although the authors do not aim to provide unexpected data to make a new point for the field, it would be nice if the authors could mention or add some implications of their analysis for the field, other than simply confirming the established phase theory. If a linguist not working on this language reads this paper, what do they gain?

The answer to comment 7 is provided in the conclusion of the paper. The authors have modified the conclusion and have also added some implications of the study. This can be demonstrated in the end of the conclusion section. 

Furthermore, the study provides theoretical implications; the findings of the study provide further support to Chomsky’s Phase Theory, where the analysis has shown that Mehri obeys the Phase Impenetrability Condition proposed in [2], which assumes that when all syntactic operations in a given phase have been achieved, the domain of that phase becomes impenetrable to any syntactic operation in the syntax. 

Another implication is that, unlike the assumption of [4449], which states that ergativity in Aramaic dialects is unique in Semitic tradition due to the contact with ergative languages such as Kurdish and Iranian languages, this article demonstrated that the ergative morphology is invariably attested in Mehri, which belongs to the Semitic language family. It is also hoped that the syntactic findings of the study will contribute to the understanding of the syntax of wh-questions in the Semitic language family and to the existing literature on the Arabic language and its varieties as well as on other languages in the world.

Response to Reviewer #2

“First of all, the manuscript should be carefully proofread before future re-submission. Currently, it is filled with missing/wrong references to examples and figures. For example, there are more than one figures labeled figure 1, same with figure 2; on page 7 just below figure 1, there is an in-text figure reference without any numbering; on pages 14-16, there is something wrong with example referencing (it reads “Error! Reference source no found” in text several times); on page 15, there are multiple references to a non-existing example “0”. In addition to all these errors, there are also a lot of typos, e.g. the word “noble” should be “novel” in the abstract, the word order in the gloss of example (4) is wrong, there is a sentence towards the middle of page 18: “… remains in-situ in.” which is either unfinished, or contains an extra “in”, etc.. All these problems are extremely distracting, and make the manuscript hard to read.

The paper has been proofread. The missing/wrong references to examples and figures have been checked and corrected accordingly in the text of the paper. “The word “noble” should be “novel” in the abstract” has been corrected. “The word order in the gloss of example (4)” has been corrected. “There is a sentence towards the middle of page 18: “… remains in-situ in.” which is either unfinished, or contains an extra “in”, this sentence has been corrected in the text of the paper. 

The second reviewer stated that “on pages 14-16, there is something wrong with example referencing (it reads “Error! Reference source no found” in text several times); on page 15, there are multiple references to a non-existing example “0.”

"Error! Reference source not found" is corrected everywhere in the manuscript. 

Concerning this comment "there are multiple references to a non-existing example “0" has been corrected. 

- Section 1: This is a rather lengthy discussion on wh-movement in Arabic. This whole discussion has very little to do with how wh-movement works in Mehri, which I assume is the main point of this paper. Just because Arabic and Mehri are related languages does not mean that the authors can make syntactic assumptions about Mehri based on what has been established in Arabic. I recommend either removing this discussion entirely, or make clear what this discussion has to do with the rest of the paper.

The answer to comment 1 is provided in the text of the paper. The authors have made clear what this discussion on wh-movement in Arabic has to do with the rest of the paper.

They have provided some justifications. Please see the beginning of that section on wh-movement in Arabic.

This section reviews the previous analyses conducted by Arab scholars on wh-questions in Arabic syntax. The reasons why this section begins reviewing the previous analysis of wh-questions in Arabic can be attributed to the following: (i) both Mehri and Arabic belong to the Semitic language family, which also includes Aramaic, Amharic, Syriac, Canaanite, and southern Arabian languages- one of which is Mehri [13]. (ii) Both Mehri and Arabic exhibit two prominent word orders: SVO and VSO. Apart from Arabic that exhibits agreement asymmetry between a verb and its post-verbal subject within VSO order, Mehri appears with a full agreement in both SVO and VSO word orders [12]. (iii) Both Mehri and Arabic exhibit certain syntactic properties with regard to wh-movement in unaccusative and (un)ergative structures. The analysis of Arabic wh-questions reviewed in this subsection will facilitate the understanding of the syntactic discussion of wh-questions in Mehri’s unaccusative and (un)ergative structures. Moreover, it is true that Arabic and Mehri indeed are related languages, but it does not mean that the syntactic assumptions in the present study about Mehri will be based on what has been established in Arabic. What distinguishes the analysis adopted in this study from those on Arabic wh-questions reviewed below is that the analysis about Mehri is based on the Phase Theory of Chomsky (2008)[2], whereas the previous accounts on Arabic wh-question discussed below adopted the Government and Binding theory and early minimalist frameworks. Consequently, the approaches of the analyses are different and the findings are dissimilar.

- p.12: I recommend having a tree demonstrating the split-CP hypothesis, which is implicitly adopted throughout the manuscript (the ForceP-FocP-TopP left periphery). This is worth making clear, since the ordering of FocP and TopP is doing a lot of the heavy-lifting in accounting for examples like (11) and (12) later on.

The authors have added a sub-section on Rizzi’s (1997) Split-CP Hypothesis illustrated by Rizz’s Tree-diagram immediately after the Phase Theory framework section. 

Rizzi’s [35] Split-CP Hypothesis

In this sub-section, we introduce Rizzi’s [35] seminal analysis of the fine structure of CP. The objective of introducing Rizzi’s Split-CP is that parts of the syntactic analysis of the present study adopted Rizzi’s proposal, especially TopP and FocP projections.

A closer look at the traditional understanding of the clause reveals that it consists of three syntactic layers: CP, IP, and VP. As the CP indicates a speaker’s attitude, the IP illustrates grammatical input (tense and agreement features). The VP, on the other hand, contains arguments. Each of these layers has been split into separate projections; the CP is split in [35], the IP is in [36], and the VP in [37]. In the last two decades, [35], [38], and [39], to name a few, proposed a syntactic expansion of a functional projection in order to accommodate the complementizer (C) and other syntactically relevant material existing on the clause left periphery. [35] suggested that the CP projection of a clause structure should be split into a number of separate functional projections in the syntax in order to host elements that surface on the clause's left periphery. In Rizzi’s [35] analysis, a head complementizer (C) was viewed as a Force marker, thus heading Force Phrases. Furthermore, topicalized and focused constituents were viewed as different Topic Phrases (headed by Top) and Focus Phrases (headed by Foc). Hence, the outcome of the proposal is that there are separate projections, each one having a specifier position and a head-on clause structure. Rizzi’s Split-CP structure can be illustrated below in Figure 4.

Figure 4. Tree structure of Rizziʻs [35] Split-CP Hypothesis 

A closer look at Rizzi’s hierarchical ordering of the expanded functional projections reveals that the projection ForceP dominates TopP, which dominates FocP, which dominates FinP; this would look like (… Force … (TopP) … (Foc) … FinIP). Moreover, [38] modified the preceding proposal, where he added IntP (interrogative projection) to the clause structure to account for wh-questions. It has been demonstrated in the existing literature that in Germanic languages topics undergo V-movement, whereas they do not in English; in the latter (English), the head of the Topic does not need to be filled in the syntax. 

- Middle of p.13: “Chomsky stressed that when all syntactic operations in a given phase have been completed…” Merger of the phase head triggers the spell out of phasal complement. Merger of phase head is crucially different from “when all syntactic operations in a phase are completed”.

These two clauses have been deleted. 

- Bottom of p.17: “Meanwhile, the lexical V ‘exploded’ raises to the head T position in order to …” You haven’t yet established whether/how V-to-T head movement in Mehri works (observing that it happens in Arabic is not a justification that the same happens in Mehri). Please provide justification that V-to-T movement happens in this case, or state clearly that this is an assumption you are making (rather than stating it as part of your proposal).

In Mehri language, all lexical verbs must combine temporal and agreement features. These features are valued through embedded affixes within a verb, or sometimes through melodies as demonstrated in the lexical V fataṩṩ ‘exploded’. In contrast to the V fataṩūt ‘exploded’ that combines perfective and agreement (i.e., 3fs) features, the lexical V fataṩṩ ‘exploded’ semantically bears perfective and agreement (i.e., 3fp) features. T, on the other hand, bears the same features but they are unvalued. It, then, probes down searching for a goal or a checker. The only goal available is the lexical V fataṩṩ ‘exploded’. It covertly joins T to constitute a verbal complex (T+V).

- Bottom of p. 17: “… it can be observed in the preceding examples (8) and (9) that the head T contains unvalued agreement features and an EPP feature.” How do (8) and (9) show you that T has an EPP feature? (8) and (9) only shows that the subject DP precedes the V. How do you know that the subject DPs are in Spec-TP (as opposed to, for example, Spec-TopicP)?

The finite declarative/interrogative force of C inherits formal features (i.e., agreement, temporal and case) to T. To verify these features, T probes down searching within VP for a goal. This goal should be aśxūf ‘the milk’ in (7) and anīʕāytan ḏa-ḥāyabīt ‘the camel’s breasts’ in (8-9). In (7), the external subject aśxūf ‘the milk’ must agree with a hybrid complex accusative V fiṩṩ ‘explode’ in agreement features (i.e., masculine, person & 3rd number) while the external subject anīʕāytan ḏa-ḥāyabīt ‘the camel’s breasts’ in (8-9) must agree with a hybrid complex unaccusative V fataṩṩ ‘explode’ in agreement features (i.e., feminine, plural & 3rd number) (see alqumairi and Shuib 2018). On the other hand, the internal object anīʕāytan ḏa-ḥāyabīt ‘the camel’s breasts’ in (7) covertly bears an accusative case feature. After triggering movement to Spec-TP in (8-9), the extracted unreal subject changes its original case to a nominative. This actually occurs via an Extended Projection Principle (EPP) of T that requires a subject of TP. Let us consider the following examples to illustrate the point.

10. fataṩṩ man aśxūf. 

 [explode-3fp.Perf] from the-milk

 ‘They exploded from the milk.’ 

11. aśxūf fiṩṩaysan.

 The milk explode-3ms-them 

 ‘The milk exploded them.’ 

Evidently, Mehri is a pro-drop language. In (10), the external subject is lexically dropped while its formal features within T are still associated with unaccusative V fataṩṩ ‘explode’. In contrast, the clitic -aysan in (11) indicates an internal accusative object anīʕāytan ḏa-ḥāyabīt.

- More generally, how do we know that (9) is an instance of overt wh-movement, rather than topicalization of an in-situ wh-PP? Or maybe just base-generation of the wh-PP at a high position? One idea is to test whether this movement is island sensitive (and if so, that rules out the base-generation possibility). You will still need some independent evidence to rule out the topicalization possibility.

The wh-PP is an adjunct that is replaced by the PP man- aśxūf. This wh-PP receives high pitch from the speaker. It is focalized and moved to Spec-FocP. As for the evidence, let us consider a parallel example:

 hibōh ķatal aśxūf?

 ‘How did the milk throw down?’

 aśxūf ķatal baḥays.

 Or baḥays, aśxūf ķatal

 ‘The milk strongly threw down’

The wh-advP baḥays ‘strongly’ is focalized and moved to the Spec-FocP configuration. 

- p. 18: “From Mehri examples (8-9), the head T inherits tense, case, and edge features from C…” Please be clear how the two sentences in (8) and (9) show this (especially, that T inherits EF from C).

The finite declarative/interrogative force of C inherits formal features (i.e., agreement, temporal and case) to T. To verify these features, T probes down searching within VP for a goal. This goal should be aśxūf ‘the milk’ in (7) and anīʕāytan ḏa-ḥāyabīt ‘the camel’s breasts’ in (8-9). In (7), the external subject aśxūf ‘the milk’ must agree with a hybrid complex accusative V fiṩṩ ‘explode’ in agreement features (i.e., masculine, person & 3rd number), while the external subject anīʕāytan ḏa-ḥāyabīt ‘the camel’s breasts’ in (8-9) must agree with a hybrid complex unaccusative V fataṩṩ ‘explode’ in agreement features (i.e., feminine, plural & 3rd number) (see Alqumairi and Shuib 2018). On the other hand, the internal object anīʕāytan ḏa-ḥāyabīt ‘the camel’s breasts’ in (7) covertly bears an accusative case feature. After triggering movement to Spec-TP in (8-9), the extracted unreal subject changes its original case to a nominative. This actually occurs via an Extended Projection Principle (EPP) of T that requires a subject of TP. To illustrate the point let us consider the following examples:

10. fataṩṩ man aśxūf. 

 [explode.3fp.Perf] from the milk

 ‘They exploded from the milk.’ 

11. aśxūf fiṩṩaysan

 The.milk explode.3ms-them 

 ‘The milk exploded them.’ 

Evidently, Mehri is a pro-drop language. In (10), the external subject is lexically dropped while its formal features of within T are still associated with unaccusative V fataṩṩ ‘explode’. In contrast, the clitic -aysan in (11) indicates an internal accusative object anīʕāytan ḏa-ḥāyabīt ‘the camel’s breasts.’

- Top of p.19 “Since this DP is definite and is located in Spec-TopP, it can be assumed as a Topic”. Why is it in Spec-TopP, as opposed to Spec-TP? No independent evidence is provided. Some diagnostics for structural height could be helpful here (e.g. ordering between the DP and certain TP-adjuncts). - bottom of p.19: “… it is triggered to move to Spec-TopP because it is definite.” Why would definiteness trigger movement to Spec-TopP in Mehri? Right now the wording makes it sounds like all definite DPs in Mehri need to undergo Topic movement (which can’t be right). Is there independent evidence for this?

According to Watson (2012) and Rubin (2010), Mehri exhibits (in)definiteness. The definite markers in Mehri are always prefixed by a- and ha- such as, arawram ‘the sea’ rawram ‘a sea’, hayabīt ‘the camel’ yabīt ‘a camel’ yabīt ‘a camel’, while the indefinite marker is null. Semantically, with the definite articles, the speakers always signal to the hearer that the referent of the DPs is familiar to both interlocutors. However, let us consider the following examples:

A). anīʕāytan ḏa-hayabīt lyakmah fataṩṩ man aśxūf.

 The camel’s breasts(def.3pf.) those(3pf) explode(erg.perf.3pf.) from the-milk

 ‘Those camel’s breasts exploded from the milk.’

B). nīʕāytan ḏa-yabīt fataṩṩ man aśxūf. 

 camel’s breasts (indef. 3pf.) explode(erg.perf.3pf.) from the-milk

 ‘Camel’s breasts exploded from the milk.’

C). *nīʕāytan ḏa-yabīt lyakmah fataṩṩ man aśxūf.

 camel’s breasts(indef.3pf.) those(3pf) explode(erg.perf.3pf.) from the-milk

The example in (A) shows the situational use of the definiteness. The referent of the DP anīʕāytan ḏa-hayabīt “the camel’s breasts” is familiar to the interlocutors through a physical situation, which contributes this familiarity. In other words, the hearer can topicalize and easily pick out the intended referent because it is familiar to him from an immediate and visible situation. In (B), however, we have an example of unfamiliar use of the indefinite DP nīʕāytan ḏa-yabīt ‘camel’s breasts’, its referent is not familiar to the hearer from a visible situation. In contrast, the example in (C) is not acceptable because the indefinite DP cannot be associated with a definite demonstrative article such as, lyakmah ‘those.’ Let us consider the following tree-diagram to illustrate the point. 

As demonstrated in the above clause structure of Mehri, the DP anīʕāytan ḏa-hayabīt lyakmah ‘those camel’s breasts’ is gradually moved from the Spec-VP projection to Spec-TP and then to Spec-TopP. 

- Is there independent evidence that wh-DPs can be licensed in-situ in Mehri?

The evidence that wh-DPs can be licensed in-situ in Mehri can be demonstrated in the following example and its clause structure. 

 haśan laykmah fataṩṩ man aśxūf?

 what those(3pf) explode (erg. perf. 3pf.) from the-milk

 ‘What (those) exploded from the milk.’

From the example above and its syntactic representation in the clause structure, the topicalized DP anīʕāytan ḏa-hayabīt ‘the camel’s breasts’ is replaced by wh-word haśan ‘what’ and is moved to the Spec-ForceP configuration, while the demonstrative article laykmah ‘those’ occupies Spec-TopP. 

Moreover, in Mehri language, two types of questions can be licensed: fronted wh-questions and in-situ wh-questions. According the existence of formal features (i.e., EF, Topic and Focus) within a clause, the following syntactic options will be constructed: 

1. When the external DPs are definite, the wh-subject will move tp the spec-TopP configuration.

2. When the external DPs are indefinite, the wh-subject will remain in-situ in the Spec-TP configuration. 

3. When the internal object or adjunct can be moved to the Spec-FocP configuration via an edge feature inherited from C (i.e., Force), otherwise it will remain in-situ. 

- p.21: traditionally, experiencer arguments are thought to be VP-internal arguments (Landau 2010). Under this view, wouldn’t “become tired” actually be projecting “camel” VP-internally unlike an unergative verb? Why not just use clearly unergative verbs like “walk” or “swim” in your analysis instead?

Similar to various unergative verbs (i.e., sōbaḥ ‘swim’, baķōṩ ‘run’, šawkōf ‘sleep’ etc.), the unergative verb tašaḥrawd ‘become tired’ only projects one argument. This argument must be a pre-verbal or post-verbal subject that theta-marks an experiencer or agent. We use the ša-type verbs such as tašaḥrawd ‘become tired; 3fs’ or šaḥrawd ‘become tired; 3ms’ only to represent the derived unergative verbs in Mehri language.

- middle of p.25 “… because the temporal complex is banned from crossing over the wh-word…” Why is V-to-T movement blocked by the wh-word? There is no explanation provided.

V-to-T movement is blocked by the wh-word because the wh-word replaces the external subject ḥāybīt ḥōfay ‘the milked camel’. 

- Is the unergative counterpart of example (11) possible? Under your analysis, if the adjunct “man masyer” is moved to Spec-Foc, the wh-DP could be licensed in-situ (maybe in Spec-TopP) with only covert movement of a Q-feature. This predicts a sentence like “man masyer haṡan tašaḥrawd” to be possible.

Yes, the unergative counterpart of example (11) is possible in Mehri.

- What about multiple-wh questions (e.g. Who ran to which house)? Is that a possibility in Mehri? Either way, it would be a nice testing bed for the proposed theory.

Yes, Mehri permits wh- and multi wh-word movement. This can be illustrated in the examples below.

- mūn baķūṩ lhal haśan man bayt?

 Who run (perf.3ms.) to which of house

 ‘Who ran to which house’

- mūn katūb haśan?

 Who write(perf.3ms.) what

 ‘Who wrote what?’

- wakōh hẽt atbayk mūn sabṫaw-k?

 Why you cry (prog.2ms.) who hit (imperf.3ms.)-you

 ‘Why are you crying? Who hit you?’

- wakōh hẽt ḏašḥradk, damlak haśan?

 Why you tired(perf.2ms.) do(perf.2ms.) what

 ‘Why did you become tired, what did you do?

---

## [Decision Letter · Decision Letter 1]

24 May 2023

PONE-D-23-09811R1The Syntax of Wh-Questions in Unaccusative and (Un)Ergative Structures in Mehri Language: A Phase-Based ApproachPLOS ONE

Dear Dr. Alzubi,

Thank you for submitting your manuscript to PLOS ONE. After careful consideration, we feel that it has merit but does not fully meet PLOS ONE’s publication criteria as it currently stands. Therefore, we invite you to submit a revised version of the manuscript that addresses the points raised during the review process. Please submit your revised manuscript by Jul 08 2023 11:59PM. If you will need more time than this to complete your revisions, please reply to this message or contact the journal office at plosone@plos.org. Please include the following items when submitting your revised manuscript:A rebuttal letter that responds to each point raised by the academic editor and reviewer(s). You should upload this letter as a separate file labeled 'Response to Reviewers'.A marked-up copy of your manuscript that highlights changes made to the original version. You should upload this as a separate file labeled 'Revised Manuscript with Track Changes'.An unmarked version of your revised paper without tracked changes. You should upload this as a separate file labeled 'Manuscript'.

We look forward to receiving your revised manuscript.

Kind regards,

Thiago P. Fernandes, PhD

Academic Editor

PLOS ONE

Additional Editor Comments:

Thank you for your valuable edits.

There are a few things to be worked on (specifically when it comes to “the literature lacks of” because there are studies on the matter) & I think the authors can refine their debate.

Please carefully address reviewers’ comments, and the new one - since a previous reviewer was not able due to workload. These comments are really interesting too, so both reviewers’ criticisms need to be considered.

Reviewers' comments:

Reviewer's Responses to Questions

**Comments to the Author**

1. If the authors have adequately addressed your comments raised in a previous round of review and you feel that this manuscript is now acceptable for publication, you may indicate that here to bypass the “Comments to the Author” section, enter your conflict of interest statement in the “Confidential to Editor” section, and submit your "Accept" recommendation.

Reviewer #2: All comments have been addressed

Reviewer #3: (No Response)

2. Is the manuscript technically sound, and do the data support the conclusions?

Reviewer #2: Yes

Reviewer #3: Partly

3. Has the statistical analysis been performed appropriately and rigorously? 

Reviewer #2: N/A

Reviewer #3: N/A

4. Have the authors made all data underlying the findings in their manuscript fully available?

Reviewer #2: Yes

Reviewer #3: Yes

5. Is the manuscript presented in an intelligible fashion and written in standard English?

Reviewer #2: (No Response)

Reviewer #3: (No Response)

6. Review Comments to the Author

Reviewer #2: Thank you for the responses to my previous comments and questions.

This draft is ready for publication with some very minor revisions:

- For example, on p.12, [Spec, vP] is incorrectly spelled as [Sec, vP].

- On p.13, "d-structure" should be "D-structure", following the notational tradition in syntax.

- In Figure 4, please be clear what the stars on the two TopP projections represent. Also, Rizzi used IP instead of TP in his notation, but this manuscript uses TP throughout. Please make sure it is consistent, and maybe change the IP to TP in Figure 4 to reduce confusion by readers.

- Rizzi has two TopP projections, a high one above FocP, and a low one below FocP. You are only making use of the high one (as shown in Figure 8). Please explicitly state this choice, and provide motivations if you can.

- On p.17, the "FinIP" in "(… Force … (TopP) … (Foc) … FinIP)" should be FinP

- Sometimes the authors use the notation [Spec, XP], and sometimes they use [Spec-XP]. Please be consistent.

- Bottom of p.30: v*p should be v*P

Reviewer #3: This paper deals with an interesting topic on wh-questions in Mehri unaccusative and unergative constructions. However, the quality of this paper is negatively influenced by the organization of this paper and the (in-)completeness of data. It still requires substantial revision to meet the publication standard, for the following reasons:

First, much work should be done to improve the logical arrangement of the contents: (1) As a journal paper, the research question(s) should be presented clearly in the beginning, say, in the introduction part or a separate part before data analysis. However, the introduction part is devoted to a detailed description of the Melhi language, without touching upon the research questions per se. (2) Without clearly presenting the research questions, the authors conduct the literature review in a rather vague manner, ranging from wh-constructions in Arabic to the general distinction between unergatives and unaccusatives. Some of the reviews, however, are not closely related to the research questions. It may work for theses, but not for journal articles. (3) Since the research questions are not clear, the conclusions are far from being concise and convincing enough.

Second, I would like to read the answers to the following questions:

(1) What is the working mechanism for wh-questions in active constructions? Is it wh-movement to [Spec, CP] or wh-in-situ? And why?

(2) What are the patterns for wh-questions in unaccusatives? Do they follow wh-movement or wh-in-situ? And why?

(3) What are the patterns for wh-questions in unergatives? Do they follow wh-movement or wh-in-situ? And why?

(4) What are the main differences with regard to wh-questions in the unaccudatives and in the unergatives? Which one patterns more with wh-questions in active constructions? Why?

Third, in order to work out the patterns for wh-questions, I found the data provided are far from being adequate, for example:

(1) In the transitive constructions (pp5-12), the authors provided six examples. As for the wh-questions for objects, is it possible to resort to the wh-in-situ mechanism? If yes, are there any semantic differences (e.g., focus reading vs. non-focus reading) between wh-movement and wh-in-situ, given that FocP is crucial in the analysis of wh-movement? As for the wh-questions for subjects, is it possible to have wh-in-situ in the VSO patterns? I suggest that a full pattern of wh-questions for transitive constructions be well presented, before moving to wh-questions in unaccusatives and unergatives.

(2) In the unaccusatives, [PP + V + WHsubj] (see example 14 on page 23) is not acceptable. However, we don’t know whether [PP + V + Subject] is acceptable. Without this information, I can hardly judge whether the authors’ analysis is on the right track or not.

(3) In the unergatives, [V + WHsubj + PP] (see example 18 on page 28) is not acceptable. How about [V + WHsubj + Object] in transitive constructions?

Without adequate examples and clear patterns, the explanations cannot be sufficiently convincing.

Fourth, I have some questions regarding the following details:

(1) The accusative verb (on page 19) � Do you mean a “causative verb”?

(2) What is the “illogical nominative subject” (p 25)? Why is it illogical? Any references?

7. PLOS authors have the option to publish the peer review history of their article (what does this mean?). If published, this will include your full peer review and any attached files.

Reviewer #2: **Yes: **Jiayi Lu

Reviewer #3: No

---

## [Author Response · Author response to Decision Letter 1]

5 Jun 2023

Response to reviewers has been attached.

---

## [Editor Report · Decision Letter 2]

23 Jun 2023

PONE-D-23-09811R2The Syntax of Wh-Questions in Unaccusative and (Un)Ergative Structures in Mehri Language: A Phase-Based Approach

PLOS ONE

Dear Dr. Alzubi,

Thank you for submitting your manuscript to PLOS ONE. After careful consideration, we feel that it has merit but does not fully meet PLOS ONE’s publication criteria as it currently stands. Therefore, we invite you to submit a revised version of the manuscript that addresses the points raised during the review process.

I’d like to thank authors’ efforts & and endeavour, but I think there is still a remaining concern. To speed up, but also to avoid another invitation of new reviewers, I’d highly recommend the authors to carefully check and address them.

As noticed, the authors have addressed almost everything raised by the reviewers, but some of them did not report or get back for another round. In this sense, remaining aspects (this is an overall concern) are presented here compiling the unsolved issues.

Overall, the authors need to better work on Introduction. There is a lack of in-depth debate of the rationale and background of your study. I do not think a simple paragraph would be enough to cover “what’s new” and “what the literature has been saying – on the language’s theories perspective. I am not sure if this was misinterpreted, but the Reviewer requested an extension of Introduction, not only one-two new sentences. This could be considered as a not-so-quick process. After this, the reviewer raised some questions and the authors promptly addressed them. Nevertheless, this should be placed in text (as amendments, i.e. as updated sentences or paragraphs covering these gaps; this should not be written in the same way as the authors placed to the reviewer, but as plain text). Finally, I think that would be fine if the authors could address the concerns on stars and/or nomenclature. Although I understand the authors’ point that this won’t be correct due to copyright etc. this can be placed as Sup. File with a sentence “adapted from…” or something similar. This would be really interesting for other researchers and readers too.

We look forward to receiving your revised manuscript.

Kind regards,

Thiago P. Fernandes, PhD

Academic Editor

PLOS ONE

Journal Requirements:

Additional Editor Comments:

Please read my comments.

---

## [Author Response · Author response to Decision Letter 2]

25 Jul 2023

Responses to reviewers’ Comments

PONE-D-23-09811R2

The Syntax of Wh-Questions in Unaccusative and (Un)Ergative Structures in Mehri Language: A Phase-Based Approach

PLOS ONE

Dear Dr. Alzubi,

Thank you for submitting your manuscript to PLOS ONE. After careful consideration, we feel that it has merit but does not fully meet PLOS ONE’s publication criteria as it currently stands. Therefore, we invite you to submit a revised version of the manuscript that addresses the points raised during the review process.

I’d like to thank authors’ efforts & and endeavour, but I think there is still a remaining concern. To speed up, but also to avoid another invitation of new reviewers, I’d highly recommend the authors to carefully check and address them.

As noticed, the authors have addressed almost everything raised by the reviewers, but some of them did not report or get back for another round. In this sense, remaining aspects (this is an overall concern) are presented here compiling the unsolved issues.

Responses to reviewers’ Comments

Reviewers#3 

Overall, the authors need to better work on Introduction. There is a lack of in-depth debate of the rationale and background of your study. I do not think a simple paragraph would be enough to cover “what’s new” and “what the literature has been saying – on the language’s theories perspective. I am not sure if this was misinterpreted, but the Reviewer requested an extension of Introduction, not only one-two new sentences. This could be considered as a not-so-quick process.

Introduction: The Introduction has been modified, where an in-depth analysis has been added (kindly see pages 4- 8). Please have a look at the Introduction section. 

Literature Review: The authors have also added a sub-section to the literate review, which reviewed the previous studies on wh-questions in Mehri. Kindly see the first sub-section on literature review (pages 8- 11).

After this, the reviewer raised some questions and the authors promptly addressed them. Nevertheless, this should be placed in text (as amendments, i.e. as updated sentences or paragraphs covering these gaps; this should not be written in the same way as the authors placed to the reviewer, but as plain text). 

All the answers to the reviewers’ comments and questions have now been placed in the text of the paper. The authors did not a leave a single question or comment without placing its answer in the text. Please have a look at the text of the paper. There are two attached files: one shows the paper with track changes (where modifications and changes have been incorporated in the text in RED) and the other one is a fair copy of the paper after incorporating all the required changes into the text). 

Kindly see the following pages to trace the changes and modifications:

1. Pages 26 to 29

2. Page 27 (Endnote iii)

3. Page 33

4. Pages 35 to 36

5. Page 38 (Endnote iv)

6. Pages 38 to 40

Also, two endnotes have been added to the text in response to the reviewers’ comments: endnote iii on page 27 and endnote iv on page 38. These endnotes are placed in the end of the paper, immediately after references (kindly see page 55). 

Reviewer #1

Finally, I think that would be fine if the authors could address the concerns on stars and/or nomenclature. Although I understand the authors’ point that this won’t be correct due to copyright etc. this can be placed as Sup. File with a sentence “adapted from…” or something similar. This would be really interesting for other researchers and readers too.

Concerning the stars and/ or nomenclature pointed out by one of the reviewers, the authors have added “adopted from (Rizzi, 1997, p. 297) to the clause structure of Rizzi on page 17, placed in the text of the paper. They have also added a Sup. File for this issue upon the request of the reviewer. This is also shown below. 

Figure 4. Tree structure of Rizzi's [1997] Split-CP Hypothesis (adopted from Rizzi, 1997, p. 297)

Note: This clause structure is on page 23 (in the text of the paper).

---

## [Editor Report · Decision Letter 3]

26 Jul 2023

The Syntax of Wh-Questions in Unaccusative and (Un)Ergative Structures in Mehri Language: A Phase-Based Approach

PONE-D-23-09811R3

Dear Dr. Alzubi,

We’re pleased to inform you that your manuscript has been judged scientifically suitable for publication and will be formally accepted for publication once it meets all outstanding technical requirements.

Kind regards,

Thiago P. Fernandes, PhD

Academic Editor

PLOS ONE

Additional Editor Comments (optional):

Thank you for your edits. Please, change vernacular terms to “study” (refer to your manuscript and the text as “study” or some similar term; i.e. “our study” , “this work”) etc. and also remove bold words, although I don’t think this will be placed in typesetting.

Wishing you success with the study.
---

## [Editor Report · Acceptance letter]

28 Jul 2023

PONE-D-23-09811R3 

The Syntax of Wh-Questions in Unaccusative and (Un)Ergative Structures in Mehri Language: A Phase-Based Approach 

Dear Dr. Alzubi:

I'm pleased to inform you that your manuscript has been deemed suitable for publication in PLOS ONE. Congratulations! Your manuscript is now with our production department. 

Kind regards, 

on behalf of

Dr. Thiago P. Fernandes 

Academic Editor

PLOS ONE